# Differential interaction patterns of opioid analgesics with μ opioid receptors correlate with ligand-specific voltage sensitivity

Sina B Kirchhofer[1,2,3], Victor Jun Yu Lim[4], Sebastian Ernst[1], Noemi Karsai[2,3], Julia G Ruland[1], Meritxell Canals[2,3], Peter Kolb[4]*, Moritz Bünemann[1]*

[1]Department of Pharmacology and Clinical Pharmacy, University of Marburg, Marburg, Germany; [2]Division of Physiology, Pharmacology and Neuroscience, School of Life Sciences, Queen's Medical Centre, University of Nottingham, Nottingham, United Kingdom; [3]Centre of Membrane Protein and Receptors, Universities of Birmingham and Nottingham, Midlands, United Kingdom; [4]Department of Pharmaceutical Chemistry, University of Marburg, Marburg, Germany

**Abstract** The μ opioid receptor (MOR) is the key target for analgesia, but the application of opioids is accompanied by several issues. There is a wide range of opioid analgesics, differing in their chemical structure and their properties of receptor activation and subsequent effects. A better understanding of ligand-receptor interactions and the resulting effects is important. Here, we calculated the respective binding poses for several opioids and analyzed interaction fingerprints between ligand and receptor. We further corroborated the interactions experimentally by cellular assays. As MOR was observed to display ligand-induced modulation of activity due to changes in membrane potential, we further analyzed the effects of voltage sensitivity on this receptor. Combining in silico and in vitro approaches, we defined discriminating interaction patterns responsible for ligand-specific voltage sensitivity and present new insights into their specific effects on activation of the MOR.

*For correspondence:
peter.kolb@uni-marburg.de (PK);
Moritz.buenemann@staff.uni-marburg.de (MB)

**Competing interest:** The authors declare that no competing interests exist.

## Editor's evaluation

This valuable study explores the interactions of different ligands with mu-opioid receptors (MORs) that differ in how membrane voltage influences their ability to modulate receptor activity. This is a relatively poorly understood phenomenon that may have unappreciated biological and clinical relevance because MORs are expressed in excitable cells where membrane voltage dynamically fluctuates. Using structure-based computational approaches and functional measurements, the authors uncover solid correlations between ligand interaction patterns with the receptor and the voltage sensitivity of its activation of the receptor. The work will be of interest to those studying the mechanism of GPCRs and the opioid receptor field in particular.

## Introduction

Opioids, agonists at the μ opioid receptor (MOR), are the most effective analgesics in clinical use. However, their pain killing effects are accompanied by severe side effects, like respiratory depression and addiction. Their high risk for abuse and overdose led to the opioid crisis in the USA with more than 80,000 deaths caused by opioid overdose in 2021 alone, on a rising trend (*CDC, 2022*).

Especially synthetic drugs, such as fentanyl, are responsible for the majority of the observed deaths. The currently used opioid analgesics differ not only in their chemical structure, but also with respect to their potency, efficacy, and kinetics to activate Gi/o proteins via MOR. Furthermore, they may exhibit differences in their efficacy to induce arrestin recruitment to MOR. There have already been attempts to develop more effective and safer opioids through a structure-based approach (*Manglik et al., 2016*; *Schmid et al., 2017*). In any case, because of the observable differences between the different opioids, it is important to understand details of ligand-receptor interactions. We recently showed that ligand-induced MOR activity is modulated by the membrane potential, and that the effect and extent of this voltage sensitivity is ligand-specific (*Ruland et al., 2020*). As the MOR is mainly expressed in highly excitable tissue and the effect of voltage modulation of MOR is present in native tissue (*Ruland et al., 2020*), the voltage sensitivity of this receptor might have a strong, still unexplored, physiological relevance, which is still neglected in the majority of studies on the MOR and GPCRs in general. As a matter of fact, since the first report of voltage sensitivity of the muscarinic $M_2$ receptor (*Ben-Chaim et al., 2003*), several other GPCRs have been observed to be modulated in their activity depending on the membrane potential. Moreover, these effects were found to be ligand-specific (*Birk et al., 2015*; *López-Serrano et al., 2020*; *Moreno-Galindo et al., 2016*; *Navarro-Polanco et al., 2011*; *Rinne et al., 2013*; *Rinne et al., 2015*), indicating that the voltage effect on GPCRs is a function of the receptor-ligand interactions. However, a general mechanism of voltage sensitivity is still elusive.

The expression of the MOR in neurons and the strongly pronounced and ligand-specific voltage effect makes this receptor an interesting candidate for further analysis of voltage sensitivity. Moreover, due to the clinical relevance of opioids, a wide range of ligands of the MOR has been described. Analysis of the interactions of these ligands with the receptor in general would give new information on molecular determinants of ligand-specific voltage sensitivity, which could then be used in the fine-tuning of safer and more effective opioids. Therefore, we analyzed the predicted binding poses of several opioids, detected key interactions and interaction groups, and correlated these with the effects voltage has on the MOR. To do so, we performed molecular docking calculations for 10 opioid ligands, including the clinically most relevant ones, and calculated interaction patterns for these ligands. Subsequently, we experimentally corroborated the predicted interactions by Förster resonance energy transfer (FRET)-based assays and by fluorescent ligand-binding competition assays in HEK293T cells. The analysis of the ligand-specific voltage sensitivity of the MOR was further performed with FRET-based functional assays under direct control of the membrane potential, revealing a correlation of the particular interaction pattern of a ligand and the specific voltage sensitivity of the MOR. Based on these observations, by means of site-directed mutagenesis, we identified receptor regions determining the effect voltage has at the MOR.

## Results

### Voltage sensitivity of the MOR is ligand-specific

Voltage sensitivity of the MOR was investigated by utilizing single-cell FRET-based assays to study G protein activity as well as recruitment of arrestin3 to the MOR under conditions of whole-cell voltage clamp. To detect the effect of voltage on G protein activity, HEK293T cells were transfected with wild-type (WT) MOR and $G\alpha_i$-mTurquoise, cpVenus-$G\gamma_2$, and $G\beta_1$ in order to monitor $G_i$ protein activity through a decrease in the FRET emission ratio (*van Unen et al., 2016*). Agonists were applied at concentrations close to the $EC_{50}$-value to avoid signal saturation. The level of maximal stimulation was determined by the application of a saturating concentration of DAMGO in all FRET recordings. Application of morphine at –90 mV induced a robust $G\alpha_i$ activation (*Figure 1A*), depolarization to +30 mV enhanced $G\alpha_i$ activation strongly, and the effect was reversible after repolarization. A similar protocol was applied to cells stimulated with methadone (*Figure 1B*) or fentanyl (*Figure 1C*). Here, however, the depolarization induced a decrease in $G\alpha_i$ activation. Voltage affected the FRET signal only when a ligand was present and MOR was expressed (*Figure 1—figure supplement 1*). Ligand dependence of the voltage sensitivity, mainly based on a change of efficacy in receptor activation, was previously additionally reported for morphine, Met-enkephalin, DAMGO, and fentanyl (*Ruland et al., 2020*). Therefore, the MOR shows a strong ligand-specific voltage sensitivity.

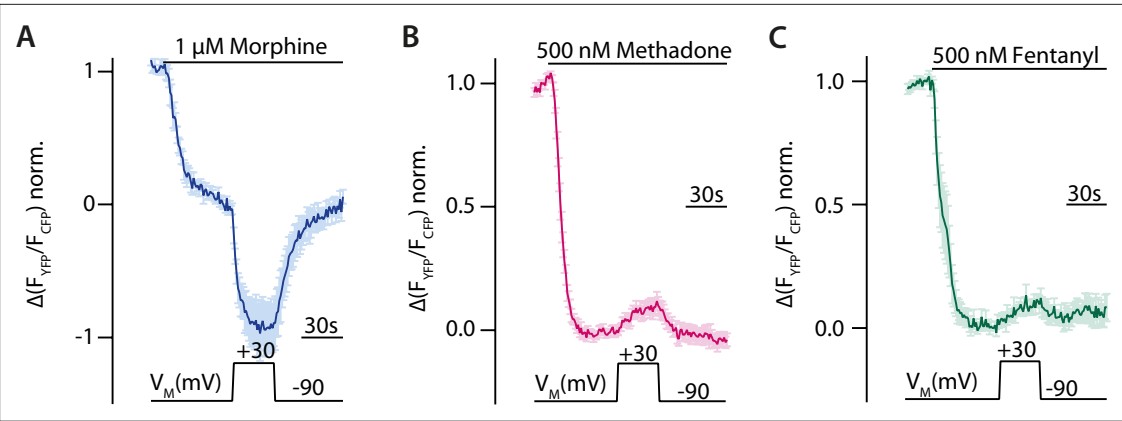

**Figure 1.** Voltage sensitivity of the μ opioid receptor (MOR) is ligand-specific. (**A–C**) Averaged Förster resonance energy transfer (FRET)-based single-cell recordings of MOR-induced Gα$_i$ activation under voltage clamp conditions with wild-type (WT) receptor, Gα$_i$-mTurquoise, cpVenus-Gγ$_2$, and Gβ in HEK293T cells are plotted for the indicated agonists (mean ± SEM; A: n=8, B: n=13, C: n=12). The applied voltage protocol is indicated below. Depolarization to +30 mV increased the morphine-induced Gα$_i$ activation (**A**) and decreased the methadone- (**B**) or fentanyl- (**C**) induced Gα$_i$ activation.

The online version of this article includes the following source data and figure supplement(s) for figure 1:

**Source data 1.** Source Data to *Figure 1A*.

**Source data 2.** Source Data to *Figure 1B*.

**Source data 3.** Source Data to *Figure 1C*.

**Figure supplement 1.** Control measurements for voltage effect of μ opioid receptor (MOR) in Gα$_i$ activation assay.

## Binding poses of different opioids at the MOR reveal distinct interaction patterns

To gain mechanistic insights into this ligand-specific voltage sensitivity, we evaluated the binding poses of several opioid ligands by molecular docking. Our docking calculations were performed based on the crystal structure of the active-state MOR (PDB: 5C1M; *Huang et al., 2015*). We decided not to use the cryo-EM structures of the MOR bound to the G protein (PDB: 6DDE and 6DDF; *Koehl et al., 2018*), as they have been solved with a peptide instead of a small molecule ligand, thus resulting in a different conformation of the orthosteric pocket. The docking calculations revealed different binding poses for the different opioids. The binding pose for morphine (*Figure 2A*) suggested D147[3.32], Y148[3.33], Y326[7.43] and the water molecules between helices 5 and 6 as important interaction partners (*Figure 2D*), and M151[3.36], V236[5.42], H297[6.52], and W293[6.48] as possible interactions, as well (numbers in superscript are according to the Ballesteros-Weinstein enumeration scheme for GPCRs; *Ballesteros and Weinstein, 1995*). In contrast, the binding pose for methadone (*Figure 2B*) indicated only a salt bridge with D147[3.32] and hydrophobic interactions and/or possible aromatic-aromatic stacking interactions with V236[5.42], H297[6.52], W293[6.48], and Y326[7.43] (*Figure 2E*). In contrast, fentanyl (*Figure 2—figure supplement 1A*) was predicted to form an H-bond with Y326[7.43] via its amide carbonyl and a salt bridge with D147[3.32] via its amide carbonyl. In addition, Q124[2.60], C217[45.50], W293[6.48], and H297[6.52] were possible interactions for fentanyl (*Figure 2—figure supplement 1B*). We further compared our fentanyl docking poses with a recently published complex structure of the MOR (PDB: 8EF5; *Zhuang et al., 2022*). Here, we found that our calculated binding pose of fentanyl (*Figure 2C*) was flipped upside down in comparison to the experimental structure, but that the overall interactions (*Figure 2F*) were comparable. This can be explained by the symmetry inherent in fentanyl, also one of the reasons why binding mode prediction for this molecule has in general been difficult. In the further analysis, we used the binding pose of fentanyl observed in the experimental structure (*Zhuang et al., 2022*). All binding poses were further investigated with a fingerprint analysis, a computational evaluation converting the interactions between a ligand and the receptor into a string of numbers, i.e., a vector. In order to reduce dimensionality, a principal component analysis (PCA) was applied to the set of fingerprints. The interactions (*Figure 2H*) that contributed strongest to the first two principal components emerged from this analysis (*Figure 2I*). On one side, interactions defining the first principal component (PC1, describing 27% of the variance observed in the interactions) were found within

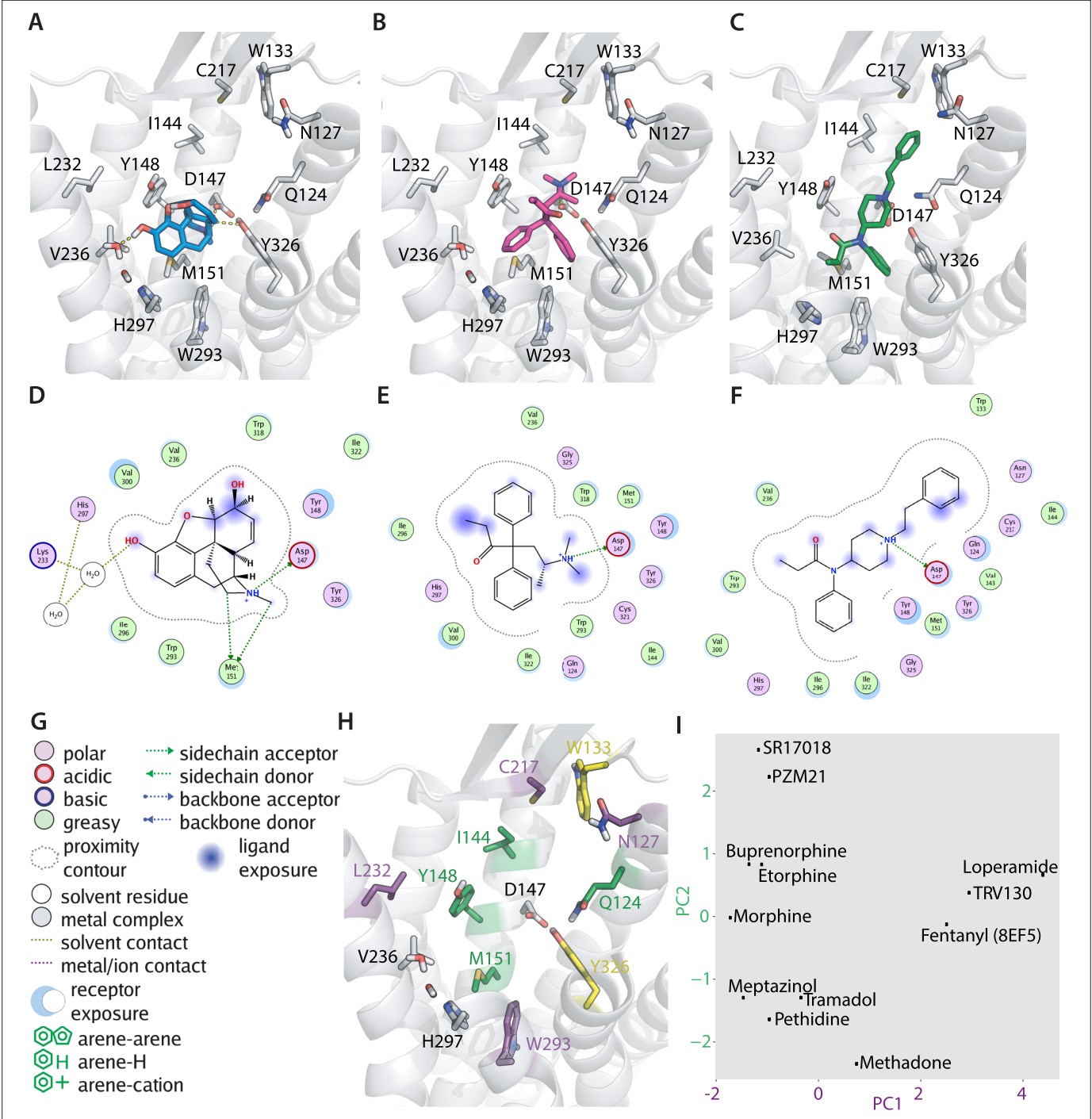

**Figure 2.** Predicted binding poses of different opioids at the μ opioid receptor (MOR) reveal differential interaction patterns. (**A–B**) Binding poses of morphine (**A**) and methadone (**B**) docked to the MOR are illustrated as a view from the extracellular side, H-bonds are indicated as dotted lines. (**C**) Binding mode of fentanyl taken from the experimental structure (PDB 8EF5). (**D–G**) 2D interaction maps displaying the calculated interactions for morphine (**D**), methadone (**E**), and fentanyl (**F**) based on the docking-derived poses shown in A–C. Key for the interaction maps is depicted in G. (**H**) Important interactions of several opioid ligands docked to MOR were identified by a fingerprint analysis, which led to the definition of the principal components plotted in (**I**). Interactions contributing strongest to component 1 (PC1) can be found within helices 2, 5, and 6 and extracellular loop 2 (N127[2.50], C217[45.50], L232[5.38], and W293[6.48], depicted in violet), whereas important interactions contributing strongest to component 2 (PC2) are mostly found in helices 2 and 3 (Q124[2.60], I144[3.29], Y148[3.33], and M151[3.36], depicted in green). Residues depicted in yellow (W133[23.50] and Y326[7.43]) are important interactions for both components. (**I**) PC1 and PC2 from the principal component analysis of the interaction fingerprints of all agonists were plotted.

The online version of this article includes the following source data and figure supplement(s) for figure 2:

*Figure 2 continued on next page*

*Figure 2 continued*

**Source data 1.** Source Data to *Figure 2I*.

**Figure supplement 1.** Binding poses of different opioids docked to μ opioid receptor (MOR).

**Figure supplement 2.** Binding poses of different opioids docked to μ opioid receptor (MOR).

helices 2, 5, and 6 and extracellular loop 2 (N127$^{2.50}$, C217$^{45.50}$, L232$^{5.38}$, and W293$^{6.48}$). On the other side, key interactions defining the second principal component (PC2, describing 15% of the variance observed in the interactions) were mostly found in helices 2 and 3 (Q124$^{2.60}$, I144$^{3.29}$, Y148$^{3.33}$, and M151$^{3.36}$). The PCA revealed diverse interaction patterns of the different opioid ligands with MOR. As a side note, the PCA plot did not change substantially when we used the fingerprint for the experimentally determined binding mode of fentanyl instead of the computational one (compare *Figure 2I* to *Figure 2—figure supplement 2A*). However, we excluded our reference agonist DAMGO from this analysis, as it is generally unfeasible to calculate a reliable binding pose of such highly flexible peptidergic ligands. Moreover, analysis of the fingerprint of the crystallographically resolved binding mode of DAMGO (*Koehl et al., 2018*) revealed a completely different interaction pattern (*Figure 2—figure supplement 2C*) in comparison to the other opioids, putting it outside of a possible applicability domain of our analysis. This is likely due to the larger size of the peptide DAMGO in comparison to the non-peptidic opioid agonists. Further, transformation of DAMGO into the already described space led to no reasonable clustering of DAMGO in comparison to the other ligands (*Figure 2—figure supplement 2D*). Along these lines, we suggest that the use of our findings in a predictive manner should only be attempted for ligands with similar physicochemical characteristics (including the size; *Supplementary file 1*) and binding locations. As the MOR-binding pocket is known to be highly flexible and pose prediction via docking could possibly be unreliable, we repeated our fingerprint analysis for all tested ligands with not only the highest ranked poses but also with the top three poses according to energy score, respectively (*Figure 2—figure supplement 2B*). The resulting fingerprints did not vary to a large extent between the top three poses, suggesting our computational pose prediction is suitable for further evaluation.

## Functional effects of site-directed mutagenesis support calculated interaction patterns of different opioids at the MOR

To experimentally corroborate the observed ligand:receptor interactions, we performed site-directed mutagenesis of several residues that were predicted to be important or not in the binding pocket of the MOR. The decision of which residue was mutated and to which amino acid was taken based on a visual investigation of the calculated binding poses. Next, we determined concentration-response curves for G protein activation in single-cell FRET measurements for the different modified receptors. To that end, we measured Gα$_i$ activation induced by MOR-WT or the mutated version of MOR at increasing concentrations of morphine, methadone, or fentanyl and compared it to the maximal activation obtained when using DAMGO. We plotted these as concentration-response curves (*Figure 3A*) and calculated pEC$_{50}$-values for each receptor variant and ligand. To further evaluate the mutants, we additionally performed fluorescent ligand competition-binding assays as described before (*Schembri et al., 2015*; *Figure 3B–C*). Therefore, we measured the displacement of the sulfo-Cy5-bearing fluorescent buprenorphine-based ligand by morphine, methadone, and fentanyl at the MOR-WT and the mutated versions of MOR and calculated pIC$_{50}$-values, where applicable. To give an overview of all mutations and their influence on Gα$_i$ activation (*Figure 3—figure supplement 1*) and competition binding (*Figure 3—figure supplements 2–3*) of the different ligands, we plotted all calculated pEC$_{50}$-values and pIC$_{50}$-values in bar graphs (*Figure 3D*). The mutation of Y148$^{3.33}$F, V236$^{5.42}$, and H297$^{6.52}$, respectively, led to a strong loss of pEC$_{50}$-value for morphine-induced Gα$_i$ activation and pIC$_{50}$-value for competition binding, indicating the importance of these residues for proper morphine binding, consistent with the docking prediction. For methadone, we identified H297$^{6.52}$ as important interaction. Furthermore, the identification of Y326$^{7.43}$ as important interaction for methadone and fentanyl was verified by Gα$_i$ activation and competition binding. Residue W293$^{6.48}$, part of the CWxP motif, which is known to be important in the activation of class A GPCRs (*Shi et al., 2002*), was identified as important interaction for both methadone and fentanyl as well. Replacement by the smaller F resulted in nearly completely abolished Gα$_i$ activation by fentanyl. In contrast, for methadone we observed

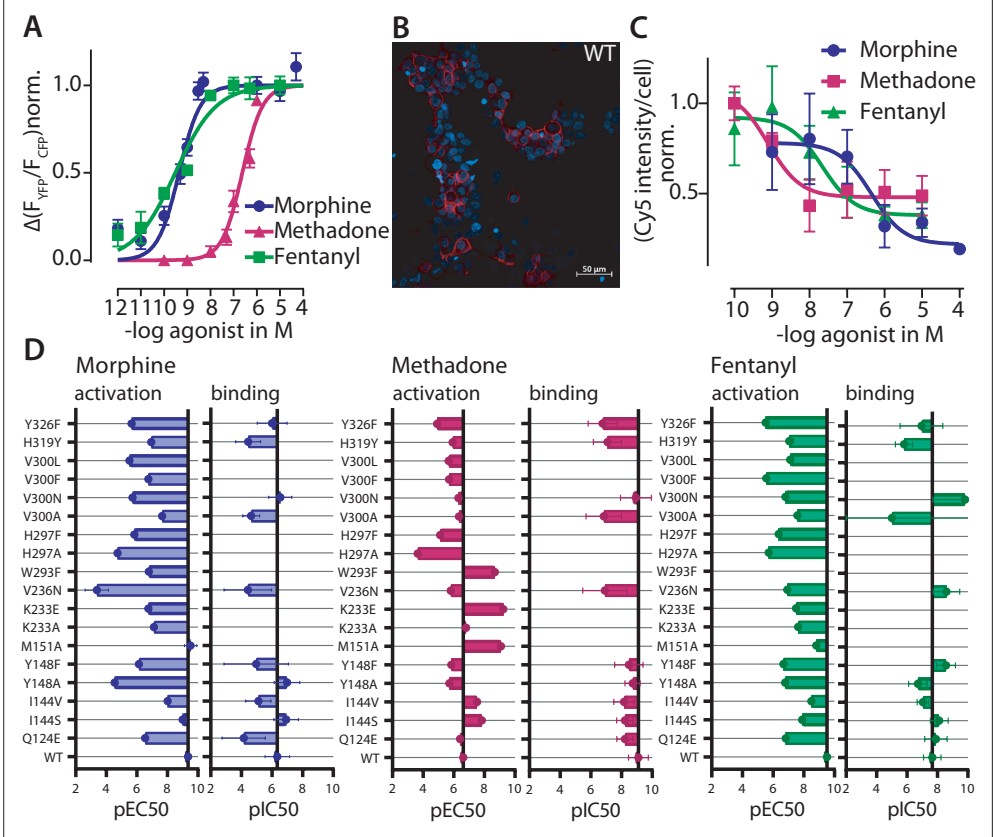

**Figure 3.** Effects on function. and ligand binding of point mutations corroborate binding poses of different opioids at the µ opioid receptor (MOR). (**A**) Concentration-response curve for $G\alpha_i$ activation induced by the depicted agonist were fitted for MOR wild-type (WT) and the $pEC_{50}$-values (morphine = 9.35, methadone = 6.62, fentanyl = 9.51) were calculated. Data was collected by single-cell Förster resonance energy transfer (FRET) measurements and each data point represents mean ± SEM. (**B**) Representative live-cell confocal image of 50 nM sulfo-Cy5-bearing fluorescent buprenorphine-based ligand (red) (*Schembri et al., 2015*) in cells expressing MOR-WT. Cells were co-stained with Hoechst33342 (blue). (**C**) Competition-binding curves for displacement of fluorescent ligand for WT MOR. Cy5 intensity was normalized to the number of cells calculated through Hoechst-staining, normalized to maximum binding and $pIC_{50}$-values (morphine = 6.35, methadone = 9.1, fentanyl = 7.66) were calculated. Each data point represents mean ± SEM of a minimum of three independent experiments performed in triplicate. (**D**) The $pEC_{50}$-values for $G\alpha_i$ activation and $pIC_{50}$-values for competition binding were plotted in a bar graph (± SEM) showing the loss or gain in $pEC_{50}$ and $pIC_{50}$ depending on the point mutation. The mutants M151A, K233A, K233E, W293F, H297A, H297F, V300F, and V300L couldn't be evaluated regarding competition binding as some mutants showed no detectable binding of the fluorescent ligand (M151A, W293F, H297A, H297F, V300F, and V300L) or showed no displacement of the fluorescent ligand (K233A and K233E), as shown in *Figure 3—figure supplement 2* and *Figure 3—figure supplement 3*. All calculated $pEC_{50}$ and $pIC_{50}$ values and the corresponding 95% confidence intervals are listed in *Supplementary file 2*.

The online version of this article includes the following source data and figure supplement(s) for figure 3:

**Source data 1.** Source Data to *Figure 3A*.

**Source data 2.** Source Data to *Figure 3C*.

**Source data 3.** Source Data to *Figure 3D*.

**Figure supplement 1.** Functional effects of the mutations displayed by $G\alpha_i$ activation and GRK2 interaction.

**Figure supplement 2.** Effects on fluorescent ligand binding of the mutations.

**Figure supplement 3.** Effects of the mutations on ligand binding determined by fluorescent ligand-binding competition assays.

**Figure supplement 4.** Expression levels of the receptor variants analyzed with western blot.

**Figure supplement 4—source data 1.** Source Data to *Figure 3—figure supplement 4A–C*.

*Figure 3 continued on next page*

*Figure 3 continued*

**Figure supplement 4—source data 2.** Source Data to *Figure 3—figure supplement 4D*.

**Figure supplement 4—source data 3.** Source Data to *Figure 3—figure supplement 4D*.

**Figure supplement 4—source data 4.** Source Data to *Figure 3—figure supplement 4D*.

**Figure supplement 4—source data 5.** Source Data to *Figure 3—figure supplement 4D*.

**Figure supplement 4—source data 6.** Source Data to *Figure 3—figure supplement 4D*.

**Figure supplement 4—source data 7.** Source Data to *Figure 3—figure supplement 4D*.

**Figure supplement 4—source data 8.** Source Data to *Figure 3—figure supplement 4D*.

**Figure supplement 4—source data 9.** Source Data to *Figure 3—figure supplement 4D*.

an increase in $G\alpha_i$ activation (*Figure 2—figure supplement 2J*, left shift by 2 orders of magnitude). However, this mutant was not able to bind the fluorescent ligand anymore (*Figure 3—figure supplement 2J*), making it impossible to evaluate the effect of this mutant in competition-binding assays. The same is true for the mutations of $M151^{3.35}$, $H297^{6.52}$, and $V300^{6.55}$ to F and L (*Figure 3—figure supplement 2F, J, K, L, O, P*). Interestingly, both mutants of $K233^{5.39}$ clearly bound the fluorescent ligand (*Figure 3—figure supplement 2G–H*), yet we were not able to observe displacement of the fluorescent ligand upon ligand application (*Figure 3—figure supplement 3G–H*). Overall, we see a high similarity in effects on function (shown by $G\alpha_i$ activation) and ligand binding (shown by competition of fluorescent ligand) induced by the point mutations. Just some mutations showed differing effects between binding and activation ($Y148^{3.33}A$ for morphine, $Y148^{3.33}F$, $V236^{5.42}N$, and $V300^{6.55}N$ for fentanyl). For these mutants, the binding was increased or not effected, but there was a stronger loss in activation of the G proteins. Overall, these experimental results are therefore congruent with the assumption that these residues are involved in ligand binding and/or elicitation of receptor response. We did not explicitly evaluate the influence on efficacy of receptor activation of the receptor mutants, as the normalization for such experiments was unfeasible for some of the mutants (*Figure 2—figure supplement 2S–V*). However, we analyzed the maximum Cy5 intensity per cell for each mutant and compared it to WT and non-transfected cells (*Figure 3—figure supplement 2T*). Here, only the mutants M151A, W293F, H297F, and V300F and -L resulted in a significant loss of Cy5 intensity in comparison to the WT receptor. However, we can't conclude from these results whether these mutants (M151A, W293F, H297F, and V300F and -L) have or do not have a significant impact on receptor function or expression levels, as we could not detect any fluorescent ligand binding. Indeed, the remaining mutants appear to have comparable ligand-binding levels to WT, as the Cy5 intensity was not significantly different (*Figure 3—figure supplement 2T*). Further, by testing for expression levels of every mutant by performing western blot analysis (*Figure 3—figure supplement 4*), we obtained similar expression levels as the WT receptor.

## Interaction pattern is consistent with agonist-specific voltage sensitivity of the MOR

As we saw different interaction patterns in the predicted binding poses of the opioid ligands, we examined these ligands for their voltage sensitivity by analyzing the extent and direction of the effect of depolarization on $G\alpha_i$ activation (*Figure 4—figure supplement 1*). We compared the effects between the ligands *Figure 4A*, with the response at +30 mV normalized to the response at –90 mV. For this, we applied the agonist at a suitable concentration to induce a robust and equivalent $G\alpha_i$ activation level in comparison to DAMGO (*Figure 4—figure supplement 1A*). This led to a great variance of the direction and magnitude of voltage-induced effects, depending on the opioid ligand used for stimulation of $G\alpha_i$ activation. Buprenorphine (*Figure 4—figure supplement 1A*) and pethidine (*Figure 4—figure supplement 1B*) enhanced their $G\alpha_i$ activation strongly from depolarization, comparable to morphine (*Figure 1A*). In contrast, etorphine (*Figure 4—figure supplement 1C*), DAMGO (*Figure 4—figure supplement 1D*), tramadol (*Figure 4—figure supplement 1E*), and PZM21 (*Figure 4—figure supplement 1F*) induced a slightly enhanced $G\alpha_i$ activation. SR17018 (*Figure 4—figure supplement 1G*) showed no apparent voltage-sensitive behavior. Moreover, meptazinol (*Figure 4—figure supplement 1H*), loperamide (*Figure 4—figure supplement 1I*), and TRV130 (*Figure 4—figure supplement 1J*) showed a voltage-dependent decrease in $G\alpha_i$ activation, comparable to the effect of fentanyl

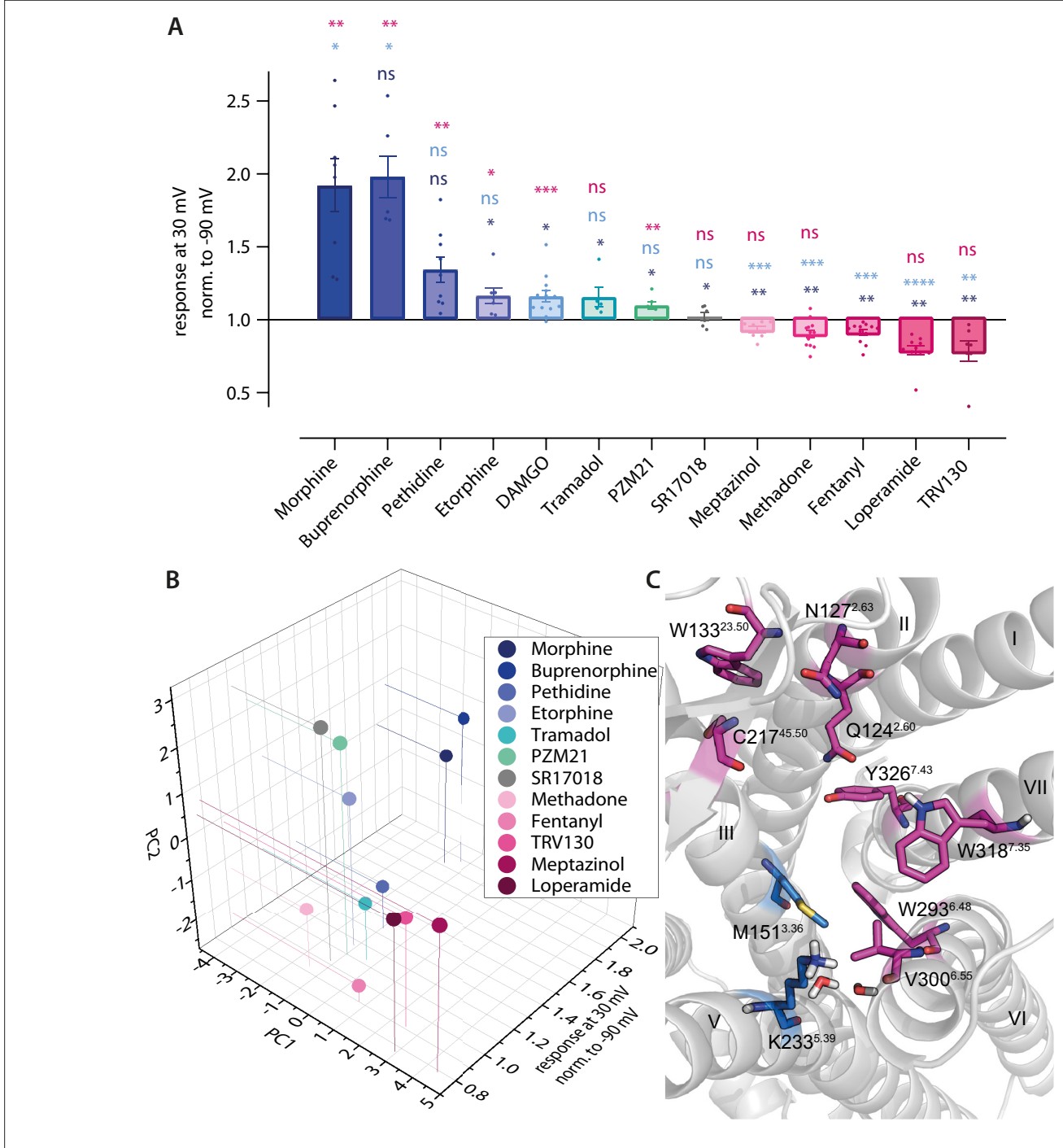

**Figure 4.** Predicted binding poses correlate with agonist-specific voltage sensitivity of μ opioid receptor (MOR). (**A**) Förster resonance energy transfer (FRET)-based single-cell recordings of Gα$_i$ activation under voltage clamp conditions induced by different opioid agonist were analyzed for agonist-specific voltage-sensitive behavior (*Figure 1* and *Figure 4—figure supplement 1A–J*). For this, the response of agonist-induced Gα$_i$ activation at +30 mV was normalized to the response at –90 mV. The applied agonist concentrations induced approximately the same Gα$_i$ activation level for all used agonists. Statistical significance was calculated compared to depolarization effect induced by morphine (dark blue), DAMGO (bright blue), and fentanyl (magenta) by an ordinary one-way ANOVA ($p<0.0001$) with Dunnett's T3 multiple comparisons test (ns $p>0.05$, *$p<0.05$, **$p<0.005$, ***$p<0.0005$). (**B**) Fingerprint analysis (*Figure 2I*) was combined with the effects voltage displayed on the agonists and plotted as 3D plot. The agonists fell into groups with a group arrangement comparable to the voltage-sensitive effect, with morphine, buprenorphine, pethidine, tramadol, and PZM21 in the group activating upon depolarization (blue and green spheres) and methadone, fentanyl, loperamide, and TRV130 deactivating upon depolarization (magenta spheres). SR17018 showed no voltage sensitivity and also showed a different binding mode compared to the other agonists

*Figure 4 continued*

(gray). (**C**) Detailed analysis of fingerprints split into groups regarding their voltage-sensitive behavior resulted in the possibility to define the main predicted interaction partners for both groups. The group showing increased activation induced by depolarization mainly interacts with helix 3 (M151$^{3.36}$) and helix 5 (K233$^{5.39}$) and the water network, depicted in blue. The group showing decreased activation induced by depolarization mainly interacts with ECL1 and -2 (W133$^{23.50}$ and C217$^{45.50}$), helix 2 (Q124$^{2.60}$ and N127$^{2.63}$), helix 6 (W293$^{6.48}$ and V300$^{6.55}$), and helix 7 (W318$^{7.35}$ and Y326$^{7.43}$), depicted in magenta.

The online version of this article includes the following source data and figure supplement(s) for figure 4:

**Source data 1.** Source Data to *Figure 4A*.

**Source data 2.** Source Data to *Figure 4B*.

**Figure supplement 1.** Agonist-specific voltage-sensitive behavior of the μ opioid receptor (MOR).

(*Figure 1C*). Thus, opioid ligands can be grouped according to their direction of voltage sensitivity. Comparing the docked poses of the opioids and their analyzed fingerprints, it becomes apparent that the voltage sensitivity of agonists is correlated to the predicted ligand-receptor interaction pattern, as defined by the fingerprint analysis (*Figure 4B*). As a control, we calculated the simple molecular descriptors for all ligands and observed no correlation with voltage sensitivity, making it highly unlikely that voltage sensitivity is determined by the properties of the ligand alone (*Figure 4—figure supplement 1K–L*). For reference, all fingerprints are shown in *Figure 4—figure supplement 1M*. Further analysis of the main interactions of the two groups of ligands resulted in the identification of distinct interaction motifs for both groups (*Figure 4C*). The ligands that showed enhanced activity upon depolarization mainly interacted with helix 3 (M151$^{3.36}$) and helix 5 (K233$^{5.39}$) and the water network between helices 5 and 6, while the ligands exhibiting a decrease in activation upon depolarization mainly interacted with ECL1 and -2 (W133$^{23.50}$ and C217$^{45.50}$), helix 2 (Q124$^{2.60}$ and N127$^{2.63}$), helix 6 (W293$^{6.48}$ and V300$^{6.55}$), and helix 7 (W318$^{7.35}$ and Y326$^{7.43}$). Overlaying this information on the binding pocket, two separate main interaction regions or motifs can be discerned (*Figure 4C*), one important for depolarization-induced activation (marked in blue) and one important for depolarization-induced deactivation (marked in pink), which correlate with the voltage-sensitive behavior of the ligand. We excluded DAMGO from this analysis as its binding pose – mainly because of its different size compared to the other ligands – resulted in a completely different fingerprint (*Figure 2—figure supplement 2C*). We further performed an association analysis by fitting a linear model of the interaction fingerprint entries of all agonists to the activation ratio upon depolarization for each interacting residue (*Figure 4—figure supplement 1M*). There it became obvious that in particular weak H-bonds with Y326$^{7.43}$ only appeared for ligands exhibiting a decrease in activation upon depolarization (*Figure 4—figure supplement 1M*). In contrast, interactions with M151$^{3.36}$ and K233$^{5.39}$ only appeared for agonists exhibiting an enhanced activity upon depolarization. The only exception here seems to be meptazinol. The fingerprint of meptazinol was comparable to compounds displaying a decrease in activation upon depolarization (*Figure 2I* and *Figure 4B*). Furthermore, for meptazinol the association analysis revealed a weak H-bond with Y326$^{7.43}$ and an interaction with K233$^{5.39}$, both interactions defining the opposite direction of voltage effect. This could possibly explain the relatively small voltage effect when applying meptazinol (*Figure 4A* and *Figure 4—figure supplement 1H*). In addition, SR17018 was the only ligand in this study which displayed no detectable voltage effect and did further not cluster with the other ligands. This could be explainable by the recent hypothesis stating this compound binds non-competitively to the MOR (*Stahl et al., 2021*).

## Altered ligand-receptor interactions influence agonist-specific voltage sensitivity of the MOR

As we already showed that site-directed mutagenesis alters ligand-induced G protein activation and binding of the ligand, we evaluated the influence of mutations of these potential ligand-receptor interactions on the agonist-specific voltage sensitivity of the MOR. Hereby we gained more information on potential molecular determinants for voltage sensitivity. Therefore, we measured mutated MOR-induced Gα$_i$ activation under voltage clamp conditions and compared this to the WT behavior. Agonists were applied in a concentration inducing comparable Gα$_i$ activation levels, which were determined respectively (*Figure 3D* and *Figure 3—figure supplement 1A–R*). The mutation of Y148$^{3.33}$ to F resulted in a reduced voltage effect of morphine (*Figure 5*, green), leading to just a

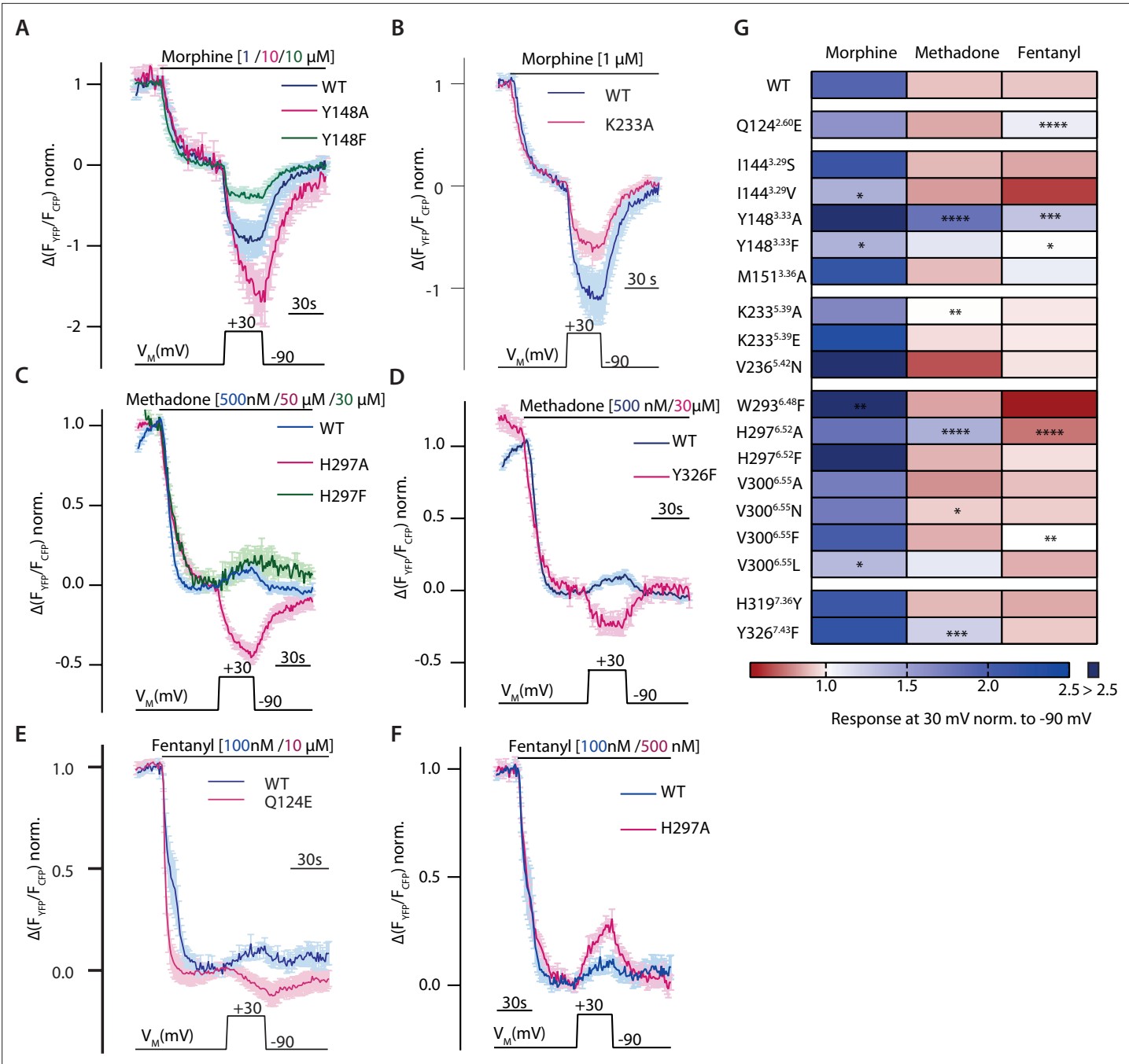

**Figure 5.** Altered ligand-receptor interactions influence agonist-specific voltage sensitivity at the μ opioid receptor (MOR). (**A–F**) Average (mean ± SEM) Förster resonance energy transfer (FRET)-based single-cell recordings of Gα$_i$ activation measured in HEK293T cells under voltage clamp conditions are plotted for the indicated agonist and mutation, with blue depicting wild-type (WT) condition and magenta or green depicting the effect of the mutant (A: MOR-WT [blue, n=8], MOR-Y148$^{3.33}$A [magenta, n=9], MOR-Y148$^{3.33}$F [green, n=6]; B: MOR-WT [blue, n=8], MOR-K233$^{5.39}$A [magenta, n=12], C: MOR-WT [blue, n=13], MOR-H297$^{6.52}$A [magenta, n=11], H297$^{6.52}$F [green, n=6]; D: MOR-WT [blue, n=13], MOR-Y326$^{7.43}$F [magenta, n=6]; E: MOR-WT [blue, n=12], MOR-Q124$^{2.60}$E [magenta, n=5]; F: MOR-WT [blue, n=12], MOR-H297$^{6.52}$A [magenta, n=10]). The applied voltage protocol is indicated below each trace. (**G**) The analyzed depolarization effects on Gα$_i$ activation induced by mutations were plotted in a heatmap regarding the applied agonist (the applied concentrations induced approximately the same Gα$_i$ activation levels for all used agonists). Response of agonist-induced Gα$_i$ activation at +30 mV was normalized to response at –90 mV, a value smaller than 1 indicates a decreased Gα$_i$ activation induced by depolarization (depicted in red), a value larger than 1 indicates an increased Gα$_i$ activation induced by depolarization (depicted in blue). Absence of a discernable voltage effect is indicated by a value around 1 (depicted in white). Significance was calculated compared to depolarization effects of the WT receptor and the respective agonist (unpaired t-test with Welch's correction [ns p>0.05, *p<0.05, **p<0.005, ***p<0.0005, ****p<0.0001]).

*Figure 5 continued on next page*

*Figure 5 continued*

The online version of this article includes the following source data and figure supplement(s) for figure 5:

**Source data 1.** Source Data to *Figure 5A*.

**Source data 2.** Source Data to *Figure 5B*.

**Source data 3.** Source Data to *Figure 5C*.

**Source data 4.** Source Data to *Figure 5D*.

**Source data 5.** Source Data to *Figure 5E*.

**Source data 6.** Source Data to *Figure 5F*.

**Source data 7.** Source Data to *Figure 5G*.

**Figure supplement 1.** Altered binding modes influence voltage sensitivity of the μ opioid receptor (MOR) activated by morphine.

slight increase of Gα$_i$ activation upon depolarization. The insertion of an A at this position instead led to a strongly increased Gα$_i$ activation, even stronger than the one for WT (*Figure 5A*, magenta). The mutation of the positively charged K233$^{5.39}$ to the neutral A reduced the voltage effect for morphine (*Figure 5B*) as well. The exchange of H297$^{6.52}$ to an A changed the direction of voltage effect for methadone (*Figure 5C*, magenta), now showing an increased Gα$_i$ activation upon depolarization. However, exchange of H297$^{6.52}$ to an F led to a voltage effect of methadone comparable to WT behavior (*Figure 5C*, green). The insertion of an F instead of Y326$^{7.43}$ changed the direction of the voltage effect for methadone (*Figure 5D*). A change of direction of voltage effect was also induced for fentanyl by the change of Q124$^{2.60}$ to an E (*Figure 5E*), now increasing Gα$_i$ activation upon depolarization. However, the mutation H297$^{6.52}$A, which inverted the voltage effect for methadone, had a divergent effect on fentanyl: here the effect of depolarization on Gα$_i$ activation led to an even stronger decrease in Gα$_i$ activation (*Figure 5F*). All effects on voltage-sensitive behavior induced by point mutations of residues involved in potential ligand-receptor interactions were plotted in a heatmap (*Figure 5G*, based on data of *Figure 5—figure supplement 1A–C*), where the agonist-induced response at +30 mV was normalized to the response at –90 mV. We did not analyze the effect of double mutants, as these displayed only weak and not evaluable Gα$_i$ activation (*Figure 5—figure supplement 1D*). Overall, although the suggested receptor interactions of morphine changed or are abolished by the mutations, depolarization increased Gα$_i$ activation in each case, albeit to a different extent. For methadone and fentanyl, the altered ligand-receptor interactions were consistent with the change in direction of the voltage effect of methadone- or fentanyl-induced Gα$_i$ activation, which was now increasing upon depolarization in nine cases. Overall, the strongest effects were induced by mutation of Y148$^{3.33}$, M151$^{3.36}$, H297$^{6.52}$, and Y326$^{7.43}$. As already shown by the fingerprint and association analysis (*Figure 4C* and *Figure 3—figure supplement 1M*), whether there's an interaction with M151$^{3.36}$ or Y326$^{7.43}$ seemed to have an influence on the direction of voltage sensitivity. Furthermore, modulation of K233$^{5.39}$, an interaction necessary for the increase in activation (*Figure 4C*, *Figure 4—figure supplement 1*), strongly diminished the voltage effect for all agonists (*Figure 5G*).

## Depolarization converts the antagonist naloxone to an agonist

Naloxone is the classical antagonist for the MOR. We analyzed the binding mode of naloxone by molecular docking, and, as the chemical structure of naloxone contains the morphinan scaffold and is highly related to morphine overall, we compared the predicted binding modes of these two ligands (*Figure 6A*). The two binding modes were highly comparable, as expected. Only the direct interaction with Y326$^{7.43}$ seems to be missing in the case of naloxone. According to the fingerprint analysis, naloxone belongs to the group of ligands that would show increased activation upon depolarization (*Figure 6B*). As it was reported before that depolarization can convert GPCR antagonists to agonists (*Gurung et al., 2008*), we also evaluated naloxone with respect to voltage sensitivity. For this, we measured MOR-induced Gα$_i$ activation under voltage clamp conditions. Application of naloxone at –90 mV induced no Gα$_i$ activation (*Figure 6C*), depolarization to +30 mV led to Gα$_i$ activation up to a level of approx. 30% of the Gα$_i$ activation induced by a saturating concentration of DAMGO. This effect was reversible after repolarization. We further analyzed this voltage effect through the application of different membrane potentials (*Figure 6—figure supplement 1A*) and fitted these to a Boltzmann function (*Figure 6D*). A comparison with the effects evoked by morphine in the same

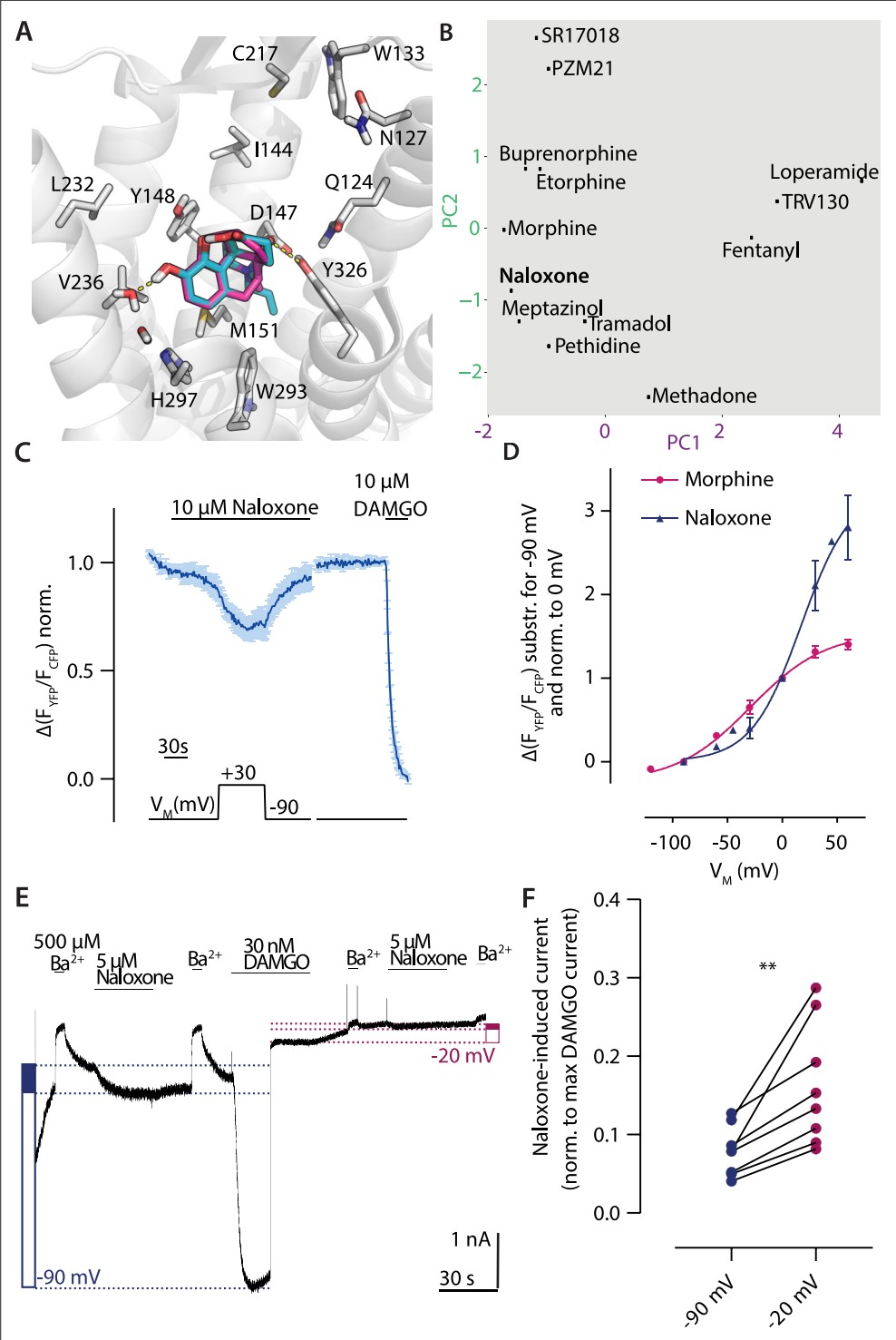

**Figure 6.** Depolarization converts the antagonist naloxone to an agonist. (**A**) Binding modes of the antagonist naloxone (cyan) compared to the agonist morphine (magenta) illustrated as in *Figure 2*. (**B**) Analyzed binding modes were plotted based on the fingerprint analysis as shown in *Figure 2*. The fingerprint of naloxone joins the group of the ligands activating upon depolarization. (**C**) Average (mean ± SEM) Förster resonance energy transfer (FRET)-based single-cell recording of μ opioid receptor (MOR)-induced Gαᵢ activation under voltage clamp conditions is plotted for naloxone with the applied voltage protocol indicated below (n=7). (**D**) Voltage dependence of naloxone (blue)-induced Gαᵢ activation was compared to morphine (magenta). The activation was determined by clamping the membrane from –90 mV to different potentials and plotted relative to 0 mV. The data

*Figure 6 continued on next page*

*Figure 6 continued*

was fitted to a Boltzmann function resulting in a z-factor of 1.17 for naloxone and 0.8 for morphine and a $V_{50}$-value of 31 mV for naloxone and –29 mV for morphine. (**E**) Representative recording of inward $K^+$ currents in HEK293T cells expressing MOR and GIRK channels, where the GIRK currents were evoked by naloxone and DAMGO. The currents were measured at –90 mV (depicted as blue dotted line) or at –20 mV (depicted as magenta dotted line). GIRK channels were blocked with 500 µM $Ba^{2+}$ as indicated. Determination of activation level induced by naloxone is indicated by the filled blue box (or magenta box, respectively) compared to the activation induced by DAMGO (empty box) (as described before; *Ruland et al., 2020*). (**F**) The GIRK current response evoked by naloxone was normalized to the maximum response evoked by DAMGO at the respective membrane potential. The responses at –90 and –20 mV were compared in the same recording, indicating an increased naloxone-induced current at –20 mV ($p < 0.05$, paired, two-tailed t-test).

The online version of this article includes the following source data and figure supplement(s) for figure 6:

**Source data 1.** Source Data to *Figure 6C*.

**Source data 2.** Source Data to *Figure 6D*.

**Source data 3.** Source Data to *Figure 6E*.

**Source data 4.** Source Data to *Figure 6F*.

**Figure supplement 1.** Depolarization converts the antagonist naloxone to an agonist.

---

setting revealed that the net charge movements upon change in membrane potential, represented as z-values, were comparable, with 1.17 for naloxone and 0.8 for morphine. Both values are also in the same range of z-values previously published for other GPCRs (*Ben-Chaim et al., 2006*; *Birk et al., 2015*; *Kurz et al., 2020*; *Navarro-Polanco et al., 2011*; *Rinne et al., 2013*). The half-maximal effective membrane potential for naloxone ($V_{50}$: +31 mV) was shifted to a more positive $V_M$ in comparison to morphine ($V_{50}$: –29 mV), indicating that the conversion of naloxone from an antagonist to an agonist requires a more positive membrane potential. We performed the identical analysis also for $G\alpha_o$ activation (*Figure 6—figure supplement 1B*), resulting in nearly identical $V_{50}$ and z-values for data fitted to a Boltzmann function (*Figure 6—figure supplement 1C*). Furthermore, we checked if this effect is also visible in assays that show no amplification. For this, we measured the direct interaction of MOR-sYFP and arrestin3-mTur2 (*Figure 5—figure supplement 1D*, see also *Ruland et al., 2020*) under voltage clamp conditions. In this case, naloxone induced no arrestin recruitment to the receptor, neither at –90 mV nor at +45 mV. This was comparable to effects of weak partial agonists like tramadol, which induced no arrestin recruitment either (*Figure 6—figure supplement 1E*). In order to further verify the observed voltage-induced conversion from antagonist to agonist for naloxone, we measured MOR-evoked inward GIRK currents at different holding potentials, as previously described (*Ruland et al., 2020*). We applied naloxone and compared the evoked $K^+$ current to a saturating concentration of DAMGO (*Figure 6E*) at –90 mV and –20 mV. The response evoked by naloxone at –90 mV was approx. 8% of the response evoked by DAMGO, whereas the response at –20 mV was approx. 16% of the response evoked by DAMGO (*Figure 6F*), indicating a significantly increased naloxone-induced current at –20 mV. To verify that the measured currents were $K^+$ currents, we applied $Ba^{2+}$ before and after every agonist or antagonist application.

All in all, this confirms the strong agonist-specific effect voltage has on the MOR, which is even able to convert antagonists to agonists. All the effects seem to be correlated with the interaction pattern of each ligand, as changes in potential important ligand-receptor interactions – either between different ligands or for one ligand in a mutant vs. the WT receptor – are correlated with the extent and direction of the voltage effect.

## Discussion

In this study, we analyzed the binding poses of several clinically relevant opioid ligands by molecular docking calculations and subsequent experimental validation of the predicted ligand-receptor interactions by FRET-based functional signaling assays, fluorescent ligand-binding studies and western blot analysis. We identified different predicted interaction patterns for morphinan ligands versus methadone and fentanyl. These differential interaction patterns were connected to ligand-specific voltage sensitivity of the MOR. Furthermore, we were able to identify important regions in the receptor which we correlated with the voltage effect on the MOR.

Specifically, our molecular docking studies and subsequent fingerprint analysis, which described the interactions between a ligand and a receptor as a vector of numbers, revealed that morphine (or agonists with the morphinan scaffold) interacted with $D147^{3.32}$, $Y148^{3.33}$, $Y326^{7.43}$ and the water networks between helices 5 and 6 as described before (*Huang et al., 2015*; *Kapoor et al., 2020*; *Lipiński et al., 2019*; *Vo et al., 2021*). Moreover, morphine displayed several interactions with helix 6, which were mostly missing for methadone and fentanyl, consistent with previous findings (*Kapoor et al., 2020*; *Lipiński et al., 2019*). The observed binding pose for methadone indicated a salt bridge with $D147^{3.32}$ as the only direct interaction, comparable to the findings of Kapoor et al. For fentanyl, we identified a salt bridge with $D147^{3.32}$ and an H-bond with $Y326^{7.43}$ as critically important interactions. Indeed, we observed a strong right shift of 4 orders of magnitude in the concentration-response curve for $G_i$ protein activation, indicating a severe loss in potency, at the tested WT-like expressing $Y326^{7.43}$ mutant, perfectly in line with our docking calculations. The same interactions could be seen in a recently published complex structure of the MOR (PDB: 8EF5; *Zhuang et al., 2022*). Although our calculated binding pose of fentanyl was flipped upside down in comparison to this experimental structure, the interactions were comparable. This can be explained by the inherent symmetry in fentanyl (*Lipiński et al., 2019*; *Qu et al., 2021*; *Vo et al., 2021*). In addition, other studies showed that there are different possible binding poses for fentanyl which can convert to each other at low energy barriers, also in line with our results (*Qu et al., 2023*). In summary, with our approach we were able to corroborate the interaction patterns calculated from the binding poses experimentally through mutagenesis.

We further evaluated several opioids regarding their voltage sensitivity by means of FRET under conditions of whole-cell voltage clamp. We identified ligands showing a strong increase in receptor activation upon depolarization of the membrane potential in a physiological range (morphine, buprenorphine, pethidine, etorphine, DAMGO, tramadol, PZM21, and naloxone). In contrast, other ligands displayed a decrease in activation (methadone, fentanyl, TRV130, loperamide, and meptazinol). Met-enkephalin (*Ruland et al., 2020*) and SR17018 displayed no apparent voltage sensitivity. This opposite direction of the voltage effect can neither be explained by the difference between partial and full agonists nor by the intrinsic ligand properties (see *Supplementary file 1*). Both partial and full agonists were included in each of the tested groups. Moreover, agonists that are hypothesized to display a bias between $G_i$ activation and arrestin recruitment compared to DAMGO (PZM21 [*Manglik et al., 2016*], TRV130 [*DeWire et al., 2013*], and SR17018 [*Schmid et al., 2017*]) were present in all groups. In conclusion, this indicated that the increased or decreased activation due to depolarization is not dependent on the degree of receptor activation. Additionally, the voltage effect was able to turn the antagonist naloxone into an agonist, comparable to the effects investigated for the $P2Y_1$ receptor (*Gurung et al., 2008*).

Importantly, we detected that the grouping of the opioids according to the direction of their voltage effect matched to a very high degree with the grouping based on the analysis of the fingerprints describing the docking-derived and experimental binding modes. These results revealed a strong ligand-specific voltage sensitivity, which seemed to be determined by the specific binding mode, and thus interaction pattern, of the ligands. Further analysis of the distinct interaction motifs of the ligand groups indicated two main interaction motifs determining the voltage effect. Helices 3 and 5 ($M151^{3.36}$ and $K233^{5.39}$) and the water network were indicated as important interaction sites for the ligands which had an activating effect upon depolarization. In contrast, a motif located mainly on helices 2, 6, and 7 ($Q124^{2.60}$, $N127^{2.63}$, $W293^{6.48}$, $V300^{6.55}$, $W318^{7.35}$, and $Y326^{7.43}$) and ECL1 and -2 ($W133^{23.50}$ and $C217^{45.50}$) appears to be important for the ligands displaying a decrease in activation. A strong influence on ligand-specific voltage sensitivity defined by differential interactions with different helices was also reported for the muscarinic acetylcholine $M_3$ receptor (*Rinne et al., 2015*). In general, there is still a lot of speculation about a possible general voltage sensing mechanism for GPCRs (*Barchad-Avitzur et al., 2016*; *Hoppe et al., 2018*; *López-Serrano et al., 2020*; *Vickery et al., 2016*). In this context, the involvement of a sodium ion bound to a conserved D was discussed (*Vickery et al., 2016*). This sodium ion seems to be important for the activation of the MOR (*Selley et al., 2000*; *Sutcliffe et al., 2017*). However, it has been shown that this sodium ion or sodium in general is not involved in the voltage sensing mechanism of GPCRs (*Ågren et al., 2018*; *Tauber and Ben Chaim, 2022*). Our approach of combining in silico and in vitro methods enabled us to identify and select important ligand-receptor interactions for each of the opioids, alter them by site-directed mutagenesis, and

test the influence of these changes on voltage sensitivity. Overall, we were not able to change the directionality of the voltage effect on MOR activation for morphinan compounds. In contrast, for methadone and fentanyl we were able to change the direction of the voltage effect following the introduction of receptor mutations. Exchange of amino acids located in helices 3 and 6 displayed the largest effects on voltage sensitivity. Especially mutation of Y148[3.33] resulted in an increased receptor activation upon depolarization for all tested ligands. A similar effect was induced by the H297[6.52]A mutation. It can be speculated that if the ligands are located closer to helix 3, the movement of helix 6, which is known to move outward upon receptor activation (*Huang et al., 2015*), could be increased upon depolarization. On the one hand, there could simply be more space for this movement if the ligands strongly interact with helix 3, further increasing the activation of the receptor. Supporting this hypothesis, we previously showed that the voltage effect induced by activation with morphine is primarily due to an increase of efficacy in receptor activation and not in affinity for the receptor (*Ruland et al., 2020*). On the other hand, ligands not showing this strong interaction with helix 3, such as fentanyl, could lose affinity for the receptor due to this movement or they might impede this movement, stabilizing the receptor in a more inactive state. Another potential base for the ligand-specific voltage effect of the MOR was presented in a recent study, where MD simulations revealed different active conformation states of the MOR depending on the bound ligand (*Qu et al., 2023*). Qu et al. found that the MOR bound to lofentanil, a derivative of fentanyl, resulted in a different conformational state than induced by the binding of another, structurally different opioid (MP). Further, DAMGO was in an equilibrium between these two possible active states, also showing the difficulty of finding a correct docking pose for this peptide. They hypothesized here that TM7 rotates in the different activation states, and especially the interaction of the residues Y326[7.43] and Q124[2.60] are crucial for these conformational changes. Interestingly, these residues displayed a strong impact on voltage sensitivity of methadone and fentanyl in our studies. One could hypothesize that these different conformational states induced by different ligands are differentially affected by voltage, resulting in an increased activity (like for morphine) or a decreased activity of the receptor (like for fentanyl).

Taken together, our results suggest that ligand-specific voltage sensitivity of MOR activation is mechanistically based on the interaction patterns between ligands and the receptor. With this study we cannot determine an accurate mechanism for the impact of voltage on the overall structure of the MOR, as the identified residues important for MOR are not known to be part of GPCR activation pathways, as described elsewhere (*Hauser et al., 2021*). However, some identified residues are to some extent part of ligand-specific conformational states of the MOR (*Qu et al., 2023*). Nevertheless, we propose that depolarization influences the conformation (or probability to reach certain conformations) of MOR in a way that increases the probability to activate receptors for ligands primarily interacting with helices 3 and 5, and the water network. Conversely, voltage decreases this probability for those ligands interacting with a motif on helices 2, 6, and 7 and the extracellular loops. These observations seem to hold true for morphinan-based ligands, but might represent a more general pattern, particularly if the influence of the ligands is considered at a helix (rather than residue) level. Indicative of the limitations of our postulates, ligands with substantially different interaction patterns, such as DAMGO, cannot be explained with our findings. As has been stated earlier in this manuscript, we suggest to limit the use of our proposed model as a predictor to ligands that have similar biophysical characteristics and binding modes as the molecules investigated here. Considering the observed ligand-specific voltage sensitivity is also seen with other receptors, it will be interesting to see if the hypothesis developed in this work also applies to those receptors as well, and maybe even to those for which voltage sensitivity has not been described yet. Our approach, strongly involving the opportunities enabled by in silico methods, allows the screening of a large number of predicted interactions and helps to choose the most information-rich receptor mutants and ligands for the subsequent in vitro analysis in a systematic and rational way. The MOR, with its diverse voltage pharmacology, was a good model system to illustrate the potential of this approach.

As MOR is expressed in neuronal tissue, which is highly excitable, a pharmacological relevance of voltage sensitivity of the MOR is very likely, albeit difficult to prove. We have already shown that the voltage sensitivity of the MOR is also reflected in brain tissue (*Ruland et al., 2020*). In this recent study we have demonstrated that the voltage modulation of MOR also affects the downstream signaling, even in a small, physiological voltage range and without overexpression of the receptor in native tissue. As it has been observed that morphine-mediated signaling is tissue-specific (*Haberstock-Debic*

*et al., 2005*), the membrane potential should be considered for the explanation of these observed effects. Further, it is known that different cell types, excitable or non-excitable, have different resting membrane potentials in a large range from –100 mV (like skeletal muscle cells) to nearly 0 mV (fertilized eggs) (*Yang and Brackenbury, 2013*). Based on this, it is intriguing that the membrane potential of these different cell types has an impact on a wide range of physiological aspects. These effects were shown among others for circadian rhythm, hearing, secretion, proliferation, cell cycle, cancer progression, and wound healing (*Abdul Kadir et al., 2018*). It seems obvious that GPCRs, known as the largest group of membrane receptors, are also highly influenced by the membrane potential and that this aspect should be considered when analyzing their signaling. So far only for muscarinic receptors, voltage insensitive mutants with otherwise WT-like agonist-binding properties have been generated and expressed in vivo. These studies revealed even a behavioral phenotype in *Drosophila* (*Rozenfeld et al., 2021*), indicating the importance of voltage sensitivity of GPCR for physiology. For the MOR, the voltage effect is only pronounced for non-endogenous opioid ligands, as the endogenous opioid met-enkephalin displayed no detectable voltage effect, as shown in *Ruland et al., 2020*, indicating a role for pharmacology rather than physiology. We suggest that the differential effect of voltage on the activity of the different opioid ligands needs to be taken into account as one possible determinant of the clinical profile of opioid drugs. A better understanding of the voltage dependence of the MOR, as achieved in our study, can potentially help with the development of safer and more effective opioids. It is, for instance, known that neurons sensing pain depolarize more often. Development of opioid ligands with a voltage dependence stronger than morphine could therefore potentially act predominantly in these depolarized cells. This would be a novel way of precise drug targeting, possibly reducing side effects, which are still the main problems of opioid therapy.

# Materials and methods
## Molecular docking and fingerprint analysis

The crystal structure of the active-state MOR (PDB code 5C1M; *Huang et al., 2015*) was prepared for docking by deletion of the N-terminus up to residue 63 and the inclusion of two water molecules (HOH 525 and HOH 546). The two water molecules were selected as they were present in both existing small-molecule-bound crystal structures (PDB codes 4DKL and 5C1M) and are involved in water bridges and hydrogen bonds with the ligand. Recent cryo-EM structures (PDB codes 6DDF and 6DDE) were not selected, as they have been solved with a peptide instead of a small molecule ligand. Using MakeReceptor (OpenEye Scientific Software, Santa Fe, NM, USA, http://www.eyesopen.com), the water molecules were defined as part of the receptor and D147[3.32] (numbers in superscript are according to the Ballesteros-Weinstein enumeration scheme for GPCRs; *Ballesteros and Weinstein, 1995*) as main interaction partner, as shown in *Surratt et al., 1994*. Ligand preparation was performed with OMEGA (OpenEye, *Hawkins et al., 2010*), using isomeric SMILES from PubChem. After docking of ligands using FRED (OpenEye, *McGann, 2011*), the best scored poses were minimized in the pocket with SZYBKI (OpenEye). Pethidine was docked a second time without the water molecules, as the pose from the initial docking was close to the side of the receptor instead of the bottom of the pocket. This is likely due to the water molecules hindering pethidine from binding at the bottom, and indeed removal of the two water molecules allowed it to reach a pose that interacted with the bottom of the pocket. The 2D ligand-protein interactions maps were generated with Molecular Operating Environment (MOE, Molecular Operating Environment, 2022.02 Chemical Computing Group ULC, Montreal, Canada) program from the binding pose. Interaction fingerprints were calculated using the program Arpeggio (*Jubb et al., 2017*), results were analyzed with PCA using scikit-learn (*Pedregosa et al., 2011*) and the first two principal components were plotted. Values on the x- and y-axis, respectively, originate from the linear combination of fingerprint features and do not carry an additional meaning, e.g., likelihood. The 10 most important interactions were determined for each component. Association analysis was performed by fitting a linear regression model of the interactions of all compounds to the activation ratio upon depolarization for each interacting residue using R programming. The F-test p-values for each interaction were computed and ranked in order to identify interactions that correlate with the activation ratio. Based on a visual investigation of the calculated binding poses we decided to

perform site-directed mutagenesis of several residues that were predicted to be important or not in the binding pocket of the MOR. Also based on this visual investigation, we decided which residue was mutated and to which amino acid.

## Plasmids

cDNAs for rat MOR-WT, MOR-sYFP2, $G\alpha_i$-YFP, $G\alpha_o$-YFP, $G\beta_1$-mTur2, $G\gamma_2$-WT, arrestin3-mTur2, GRK2-WT, GRK2-mTur2, $G\alpha_i$-WT, $G\beta_1$-WT, $G\gamma_2$-WT, bicistronic plasmid expressing GIRK3.1 and GIRK3.4 subunits and pcDNA3-eCFP have been described previously (*Ruland et al., 2020*). $G\beta_1$-2A-cpV-$G\gamma_2$-IRES-$G\alpha_{i2}$-mTur2 was purchased from Addgene (Watertown, MA, USA, plasmid #69624; *van Unen et al., 2016*). Point mutations were introduced into MOR by site-directed mutagenesis and were verified by sequencing (Eurofins Genomics, Ebersberg, Germany). The following mutagenesis primers were used (sequence 5'→3'): $Q124^{2.60}E$ agtacactgcccttgagagtgtcaactacctg; $I144^{3.29}S$ ctct gcaagatcgtgagctcaatagattactac; $I144^{3.29}V$ ctctgcaagatcgtggtctcaatagattactac; $Y148^{3.33}A$ gtgatctc aatagatgcctacaacatgttcacc; $Y148^{3.33}F$ cgtgatctcaatagatttctacaacatgttcaccag; $M151^{3.36}A$ atagattactac aacgcgttcaccagcatattc; $K233^{5.39}A$ ctgggagaacctgctcgcaatctgtgtctttatc; $K233^{5.39}E$ ctgggagaacctgctc gaaatctgtgtctttatc; $V236^{5.42}N$ cctgctcaaaatctgtaactttatcttcgctttc; $W293^{6.48}F$ gtatttatcgtctgctttaccccc atccacatc; $H297^{6.52}A$ ctgctggacccccatcgccatctacgtcatcatc, $H297^{6.52}F$ ctgctggacccccatcaagatctacgtc atcatc; $V300^{6.55}A$ cccatccacatctacgccatcatcaaagcgctg; $V300^{6.55}F$ cccatccacatctacttcatcatcaaagcg, $V300^{6.55}L$ cccatccacatctacctcatcatcaaagcg; $V300^{6.55}N$ cccatccacatctacaacatcatcaaagcgctg; $H319^{7.36}Y$ cagaccgtttcctggtacttctgcattgctttgg; $Y326^{7.43}F$ gcattgctttgggtttcacgaacagctgcctg. The mutations $Q124^{2.60}E$ (*Fowler et al., 2004*), $Y148^{3.33}F$ (*Xu et al., 1999*), $H297^{6.52}A$ (*Mansour et al., 1997*; *Spivak et al., 1997*), $H297^{6.52}F$ (*Spivak et al., 1997*), $H319^{7.36}Y$ (*Ulens et al., 2001*), and $Y326^{7.43}F$ (*Mansour et al., 1997*) have been evaluated before. Expression levels of the mutated receptor variants were comparable to the WT receptor, confirmed by western blot analysis (*Figure 3—figure supplement 4*).

## Cell culture

All experiments in this study were carried out in HEK293T cells. The used cell line was HEK tsA 201, which was a kind gift from the Lohse laboratory, University of Würzburg. Cells were cultured in high-dose DMEM supplemented with 10% FCS, 2 mM L-glutamine, 100 U/ml penicillin, and 0.1 mg/ml streptomycin at 37°C and 5% $CO_2$. Cells were transiently transfected in 6 cm Ø dishes using Effectene Transfection Reagent according to the manufacturer's instructions (QIAGEN, Hilden, Germany) 2 days before the measurement. For MOR-induced $G\alpha_i$ activation measurement, cells were transfected with 1 µg of MOR-WT or mutated MOR and 1 µg Gβ-2A-cpV-Gy2-IRES-Gai2-mTur2, for measurements of voltage dependence of morphine fitted to Boltzmann function (*Figure 6*), cells were transfected with 0.5 µg MOR-WT, 1 µg $G\alpha_i$-YFP, 0.5 µg $G\beta_1$-mTur2, and 0.25 µg $G\gamma_2$-WT. For measurement of MOR-induced GIRK currents, cells were transfected with 0.3 µg MOR-WT, 0.5 µg GIRK3.1/3.4, and 0.2 µg pcDNA3-eCFP. For MOR-induced $G\alpha_o$ activation measurement, cells were transfected with 0.5 µg of MOR-WT, 1 µg $G_o$-YFP, 0.5 $G\beta_1$-mTur, and 0.25 µg $G\gamma_2$-WT. For MOR-induced arrestin interaction, cells were transfected with each 0.7 µg of MOR-sYFP2, arrestin3-mTur2, and GRK2-WT. Cells were split on poly-L-lysine (Sigma) coated coverslips the day before the measurement. For MOR-induced GRK interaction, cells were transfected with 0.6 µg MOR-sYFP2, 0.6 µg GRK2-mTur2, 0.7 µg $G\alpha_i$-WT, 0.6 µg $G\beta_1$-WT, and 0.6 µg $G\gamma_2$-WT.

For the competition-binding experiments, HEK293T cells were cultured in high-glucose DMEM supplemented with 10% FCS at 37°C and 5% $CO_2$. Cells were transiently transfected 2 days before the measurement using PEI (PolyScience Inc, Hirschberg an der Bergstraße, Germany). Cells were sown in a concentration of 15,000 cells/well in poly-D-lysin (Sigma) coated black 96-well plate with transparent bottom (Greiner, Austria) and transfected with 100 ng DNA of MOR-WT or mutated MOR per well. The DNA:PEI ratio was 1:3 with 1 mg/ml PEI.

For the western blot experiments, HEK293T cells were cultured in high-dose DMEM supplemented with 10% FCS, 2 mM L-glutamine, 100 U/ml penicillin, and 0.1 mg/ml streptomycin at 37°C and 5% $CO_2$ and transfected 48 hr before cell lysis using PEI. Cells were sown in a concentration of 2,000,000 cells per condition in a six-well plate and transfected with 4 µg DNA of MOR-WT or mutated MOR. The DNA:PEI ratio was 1:3 with 1 mg/ml PEI.

## Reagents

DMEM, FCS, penicillin/streptomycin, L-glutamine, and trypsin-EDTA for the FRET-based and electrophysiological measurements were purchased from Capricorn Scientific (Ebsdorfergrund, Germany). DMEM, FCS, PBS, and trypsin-EDTA used for the competition-binding experiments were purchased from Sigma-Aldrich (St. Louis, MO, USA). DAMGO acetate salt, buprenorphine-HCl, fentanyl citrate, tramadol-HCl, and $BaCl_2$ were purchased from Sigma-Aldrich (St. Louis, MO, USA). Etorphine-HCl (Captivon98) was obtained from Wildlife Pharmaceuticals through Chilla CTS GmbH (Georgsmarienhütte, Germany). Loperamide-HCl was purchased from J&K Chemicals (San Jose, CA, USA), meptazinol-HCl was purchased from Biozol (Eching, Germany), morphine hydrochloride used for the FRET-based and electrophysiological measurements was purchased from Merck (Darmstadt, Germany), morphine hydrochloride used for the competition-binding experiments was purchased from Tocris (Bristol, UK) and naloxone-HCl was purchased from Cayman Chemical (Ann Arbor, MI, USA). L-methadone-HCl (used for the FRET-based and electrophysiological measurements) and pethidine-HCl were purchased from Hoechst AG (Frankfurt, Germany) and L-methadone-HCL used for the competition-binding experiments was purchased from Sigma-Aldrich (St. Louis, MO, USA). PZM21, SR17018, and TRV130 were a kind gift from Stefan Schulz and Andrea Kliewer, University of Jena, Germany (*Gillis et al., 2020*; *Miess et al., 2018*). Hoechst33342 was purchased from Thermo Scientific (Waltham, MA, USA). The sulfo-Cy5-bearing fluorescent buprenorphine-based ligand was the previously published compound 3 (2-((1E,3E,5E)-5-(1-ethyl-3,3-dimethyl-5-sulfoindolin-2-ylidene)-penta-1,3-dien-1-yl)-1-(6-((6-((6S,7R,7aR,12bS)-9-hydroxy-7-methoxy-3-methyl1,2,3,4,5,6,7,7a-octahydro-4a,7-ethano-4,12-methanobenzofuro[3,2-e]isoquinoline-6-carboxamido)hexyl)-amino)-6-oxohexyl)-3,3-dimethyl-3H-indol-1-ium-5-sulfonate,2,2,2-trifluoroacetate salt) (*Schembri et al., 2015*).

## FRET and electrophysiological measurements

Single-cell FRET measurements with or without direct control of the membrane potential were performed as described previously (*Ruland et al., 2020*). Using an inverted microscope (Axiovert 135, Zeiss) and an oil-immersion objective (A-plan 100×/1.25, Zeiss), CFP was excited by short light flashes of 430 nm (Polychrome V light source), fluorescence emission of YFP ($F_{535}$) and CFP ($F_{480}$) were detected by photodiodes (TILL Photonics Dual Emission System) with a sample frequency of 1 Hz, recording of data was performed with PatchMaster 2x65 (HEKA), and the FRET emission ratio of $F_{YFP}/F_{CFP}$ was calculated. After a necessary technical update of the setup, excitation was performed at 436 nm with a LED light source (precisExcite-100, 440 nm, CoolLED), and emission of YFP and CFP was split by an optosplit (Chroma) and detected with a CCD camera (RETIGA-R1, Teledyne Photometrics) and stored with VisiView software (Visitron Systems). As all measurements were normalized to a maximal answer within every measurement, the data was comparable between the two setup configurations. During measurements, cells were continuously superfused with either external buffer (137 mM NaCl, 5.4 mM KCl, 2 mM $CaCl_2$, 1 mM $MgCl_2$, 10 mM HEPES, pH 7.3) or external buffer containing agonist in the respective concentration using a pressurized fast-switching valve-controlled perfusion system (ALA Scientific) allowing a rapid change of solutions. For FRET measurements under direct control of the membrane potential, cells were simultaneously patched in whole-cell configuration with the membrane potential set to a defined value by an EPC-10 amplifier (HEKA). For this, borosilicitate glass capillaries with a resistance of 3–7 MΩ were filled with internal buffer solution (105 mM K⁺-aspartate, 40 mM KCl, 5 mM NaCl, 7 mM $MgCl_2$, 20 mM HEPES, 10 mM EGTA, 0.025 mM GTP, 5 mM Na⁺-ATP, pH 7.3). For measurement of GIRK currents, cells were measured in whole-cell configuration analogue to the FRET measurements in 1 kHz sampling intervals with holding potentials of –90 or –20 mV, as indicated. As inward currents were measured, the used extracellular buffer was a high K⁺ concentration containing buffer (as external buffer, but with 140 mM KCl and 2.4 mM NaCl). All measurements were performed at room temperature.

## Competition-binding experiments

Competition-binding experiments were performed as described previously (*Schembri et al., 2015*). Fluorescent ligand binding was measured in HEK293T cells 48 hr after transient transfection with WT or mutant MOR. For this, DMEM was removed, and HBSS (2 mM sodium pyruvate, 145 mM NaCl, 10 mM D-glucose, 5 mM KCl, 1 mM $MgSO_4{\times}7H_2O$, 10 mM HEPES, 1.3 mM $CaCl_2$ dihydrate, and

1.5 mM NaHCO$_3$) containing 50 nM of the sulfo-Cy5-bearing fluorescent buprenorphine-based ligand and increasing concentrations of unlabeled morphine, methadone, or fentanyl were applied and incubated for 30 min at 37°C and 5% CO$_2$. 10 min before the measurement, 1 µg/µl Hoechst33342 was added. Single-time point confocal images were captured using a Zeiss Celldiscoverer 7 LSM 900 high-content automated confocal microscope and 2 images per well were captured both using a 10× objective and the Cy5 channel (650 nm excitation, 673 emission) and the Hoechst33342 channel (348 nm excitation, 455 nm emission). All images were acquired with the same laser and optical settings.

## Western blot

For the western blots, HEK293T cells were transfected as described above. 48 hr after transfection, cells were harvested in lysis buffer (50 mM HEPES, 250 mM NaCl, 2 mM EDTA, 10% glycerol, 0.5% Igepal CA-630 [Sigma-Aldrich, Darmstadt, Germany], pH 7.5) containing Complete Mini Protease Inhibitor Cocktail (Roche Diagnostics, Penzberg, Germany), and homogenized with an Ultra-Turrax (IKA, Staufen, Germany). The extracts were centrifuged at 4°C and 10,000 × $g$ for 20 min. Supernatants were collected, and the total amount of protein determined with a Bradford assay. For western blot analysis, 40 µg of protein in 5× SDS sample buffer (312 mM Tris-HCl pH 6.8, 50% glycerol, 10% SDS, 25% β-mercaptoethanol, 0.1% bromophenol blue) were separated on an 8% SDS Gel together with peqGOLD Protein Marker V (VWR Life Science, Darmstadt, Germany) and transferred onto a PVDF membrane at 325 mA for 2.5 hr. The membranes were incubated in blocking buffer (5% fat-free dry milk powder in 1xTBST) for 2 hr at room temperature. For detecting the HA-tagged MOR, membranes were incubated overnight at 4°C with anti-HA primary antibody (1:1000, H6908, Sigma-Aldrich, Germany, RRID:AB_260070), washed 3× for 15 min with 1xTBST and incubated with HRP-conjugated anti-rabbit secondary antibody (1:3500, 7074, Cell Signaling, USA, RRID:AB_2099233) for 2 hr at room temperature. After three washing steps with 1xTBST the signals were detected using enhanced chemiluminescence detection (Thermo Fisher Scientific, Darmstadt, Germany) and the ChemiDoc XRS system (Bio-Rad Feldkirchen,Germany). For detecting the control, blots were stripped 2× for 20 min with stripping buffer (1.5% glycine, 0.1% SDS, 1% Tween 20, pH 2.2), incubated in anti-GAPDH primary antibody (1:50,000, 2118, Cell Signaling, Leiden, The Netherlands, RRID:AB_561053) overnight at 4°C and in HRP-conjugated anti-rabbit secondary antibody (1:3500, 7074, Cell Signaling, RRID:AB_2099233) for 2 hr at room temperature. The intensity of the signals was quantified with ImageJ and analyzed using GraphPad prism 8.

## Data analysis and statistics

FRET measurements were corrected for photobleaching (using OriginPro 2016) and were normalized to maximum responses within the same cell and recording. Further data analysis was performed with GraphPad Prism 8 (GraphPad Software). Data is always shown (if not indicated otherwise) as mean ± SEM and group size defined as n. Statistical analyses were performed with a paired Student's t-test or a two-tailed unpaired t-test with Welch's correction (as normality of data distribution wasn't given for every group) or, for more than two groups, by an ordinary one-way ANOVA (as SDs were significantly different, a Brown-Forsythe and Welch's ANOVA test were performed) with Dunnet's T3 multiple comparisons test, as indicated. Differences were considered as statistically significant if p≤0.05. Concentration-response curves were fitted with a non-linear least-squares fit with variable slope and a constrained top and bottom using following function:

$$Y = min + \left(X^{\text{Hill-slope}}\right) \times \left(max - min\right) / \left(X^{\text{Hill-slope}} + EC_{50}^{\text{Hill-slope}}\right)$$

where min and max are the minimal and maximal response and EC$_{50}$ is the half-maximal effective concentration. Voltage-sensitive behavior was analyzed by normalizing the answer at +30 mV (mean of last 10 s before repolarization) to the answer at –90 mV (mean of last 10 s before depolarization) with previous normalization of the whole trace to the agonist-induced answer at –90 mV as max. response. For analysis of charge movement and V$_{50}$-values, answers were subtracted from –90 mV and normalized to 0 mV. These values, now normalized to the degree of receptor activation (R) reflected by Gα$_i$ activation, were fit to a single Boltzmann function. The equation used for fitting was

$$R = R_{min} + \frac{R_{max} - R_{min}}{1 + e^{\left(\frac{V_{50} - V_M}{k}\right)}}$$

where $R_{min}$ and $R_{max}$ were the minimal and maximal response, $V_M$ the respective membrane potential, $V_{50}$ the voltage of half-maximal effect on $G\alpha_i$ activation and k the slope factor. For calculation of the z-factor, the net charge movement upon change in $V_M$ across the membrane, following equation was used:

$$z = \frac{-26}{k}$$

For analysis of GIRK current response evoked by naloxone, the responses to naloxone at either –90 or –20 mV were normalized to the max. response evoked by DAMGO at the respective $V_M$ and values generated in the same recording were compared.

Competition-binding experiments were analyzed using ZEN (blue edition) and Fiji (ImageJ). Cells stained with Hoechst33342 were counted using Fiji and the total intensity in the Cy5 channel was divided by the number of cells in the corresponding image. To fit competition-binding curves, the Cy5 intensity/cell for the increasing concentrations of agonist was normalized to the maximum Cy5 intensity/cell without competing agonist for the corresponding receptor variant. Competition-binding curves were fitted with a non-linear least-squares fit with a Hillslope of –1 using following function:

$$Y = min + (max - min) / \left(1 + 10^{((LogIC50 - X) * -1)}\right)$$

where min and max are the minimal and maximal intensity and $IC_{50}$ is the half-maximal inhibitory concentration.

## Acknowledgements

We thank Barrie Kellam and Nicholas Kindon (University of Nottingham, UK) for the provision of the fluorescent opioid ligand, Stefan Schulz and Andrea Kliewer (University of Jena, Germany) for the provision of the biased compounds PZM21, SR17018, and TRV130, and Carsten Culmsee (University of Marburg, Germany) for the provision of the antibodies. Funding for this work was provided by United Kingdom Academy of Medical Sciences Professorship Award and ONCORNET 2.0 (ONCO-genic Receptor Network of Excellence and Training 2.0) PhD training program funded by the European Commission Marie Sklodowska Curie Actions (H2020-MSCA grant agreement 860229). Open Access funding provided by the Open Access Publishing Fund of Philipps-Universität Marburg.

## Additional information

### Funding

| Funder | Grant reference number | Author |
|---|---|---|
| European Commission | H2020-MSCA- 860229 | Meritxell Canals |
| United Kingdom Academy of Medical Sciences Professorship | | Meritxell Canals |

The funders had no role in study design, data collection and interpretation, or the decision to submit the work for publication.

### Author contributions

Sina B Kirchhofer, Conceptualization, Data curation, Formal analysis, Validation, Investigation, Visualization, Methodology, Writing - original draft; Victor Jun Yu Lim, Data curation, Formal analysis, Validation, Investigation, Visualization, Writing – review and editing; Sebastian Ernst, Julia G Ruland, Data curation, Formal analysis; Noemi Karsai, Data curation, Methodology, Writing – review and editing; Meritxell Canals, Conceptualization, Supervision, Methodology, Project administration, Writing – review and editing; Peter Kolb, Conceptualization, Resources, Software, Supervision, Funding

acquisition, Validation, Investigation, Methodology, Project administration, Writing – review and editing; Moritz Bünemann, Conceptualization, Resources, Supervision, Funding acquisition, Methodology, Writing - original draft, Project administration, Writing – review and editing

### Author ORCIDs
Sina B Kirchhofer  http://orcid.org/0000-0001-6285-9054
Noemi Karsai  http://orcid.org/0009-0000-3948-4071
Peter Kolb  http://orcid.org/0000-0003-4089-614X
Moritz Bünemann  http://orcid.org/0000-0002-2259-4378

### Decision letter and Author response
Decision letter https://doi.org/10.7554/eLife.91291.sa1
Author response https://doi.org/10.7554/eLife.91291.sa2

---

## Additional files

### Supplementary files
• Supplementary file 1. Ligand properties; 2D structures were taken from Wikipedia.
• Supplementary file 2. Calculated pEC50 values for G protein activation and pIC50 values for fluorescent ligand-binding competition.
• MDAR checklist

### Data availability
All data generated or analysed during this study are included in the manuscript and supporting files; Source Data files have been provided for *Figures 1, 3–6*. Primer sequences are detailed in Materials and methods section. The full set of fingerprints can be found in *Figure 2—source data 1*.

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
