## [Editor Report]

This valuable study explores the interactions of different ligands with mu-opioid receptors (MORs) that differ in how membrane voltage influences their ability to modulate receptor activity. This is a relatively poorly understood phenomenon that may have unappreciated biological and clinical relevance because MORs are expressed in excitable cells where membrane voltage dynamically fluctuates. Using structure-based computational approaches and functional measurements, the authors uncover solid correlations between ligand interaction patterns with the receptor and the voltage sensitivity of its activation of the receptor. The work will be of interest to those studying the mechanism of GPCRs and the opioid receptor field in particular.

---

## [Decision Letter]

**Decision letter after peer review:**

[Editors’ note: the authors submitted for reconsideration following the decision after peer review. What follows is the decision letter after the first round of review.]

Thank you for submitting the paper "Differential recognition of opioid analgesics by µ opioid receptors: Predicted interaction patterns correlate with ligand-specific voltage sensitivity" for consideration by *eLife*. Your article has been reviewed by 3 peer reviewers, and the evaluation has been overseen by a Reviewing Editor and a Senior Editor. The following individuals involved in review of your submission have agreed to reveal their identity: Aashish Manglik (Reviewer #2); Xiaojing Cong (Reviewer #3).

Comments to the Authors:

We are sorry to say that, after consultation with the reviewers, we have decided that this work will not be considered further for publication by *eLife*.

Specifically, all three reviewers raised major concerns about the inherent uncertainty of the docking poses and the fingerprint analysis, which are central to the manuscript and its conclusions. While there may be some paths toward addressing this in principle (see comments below), the reviewers felt that the scope of needed work is beyond what would be reasonable for revision timeline. As a result, we cannot proceed with the current manuscript.

*Reviewer #2 (Recommendations for the authors):*

Over the past decade or so, a number of studies have demonstrated that multiple GPCRs are sensitive to voltage – i.e. their signaling properties change as a function of membrane voltage. The physiological consequence and relevance of this voltage sensitivity remains poorly understood. The authors recently reported that voltage can influence activation of the mu-opioid receptor, and that this effect is different between opioid receptor agonists (Ruland, BJP 2020), with some agonists causing increased activation of the receptor with depolarization, while other agonists (fentanyl) cause decreased activation with depolarization. In the present study, the authors aim to connect this observation to specific interactions made by different opioid receptor ligands to try to map specific contacts between ligands and the receptor that are responsible for this voltage sensitivity. Here, the authors first establish that different ligands acting at the opioid receptor are differentially sensitive to membrane voltage, e.g. the effect of morphine is increased while that of methadone and fentanyl is decreased. Based on these findings, the authors then dock various opioid ligands into the active conformation of MOR to identify differences in contacts between the ligands. A fingerprint analysis of contacts is used to cluster ligands with similar/different contacts; some of these differential interactions are probed/validated by dose response assays. The authors then suggest that grouping ligands with the fingerprint analysis approach seemed to correlate with voltage sensitivity, and that this correlation came with sufficient power to predict how meptazinol binds and the effect of voltage on etorphine. Specific mutations in the MOR binding pocket were then assessed for their effect on MOR voltage sensitivity to three prototypical opioids (morphien, methadone, and fentanyl) – here, the authors found that some mutations could reverse the directionality of voltage sensitivity for fentanyl and methadone. Finally, membrane depolarization could convert the antagonist naloxone into a weak partial agonist.

These studies convincingly show that voltage does not simply change the conformation of the MOR, leading to more or less G protein signaling activity. Instead, it is the specific interactions of a ligand with the MOR binding pocket that lead to a specific outcome – these are clearly influenced by specific ligands, and the specific contacts that a given ligand makes with the MOR binding pocket. The most exciting data to the latter point is the remarkable effect that specific mutations can have in converting the effect of voltage on methadone in Figure 6C or in the effect on naloxone.

Less clear is how well the authors can define these specific interactions. It is not clear whether the docking studies are truly informative in this regard. First, but it is not completely clear whether binding poses for various ligands are truly accurate – the authors efforts to dock fentanyl suggest that there are many potential challenges in obtaining accurate poses. Even with accurate poses, it is unclear whether the fingerprint analysis is robust. For example, it is likely the case that D147(3.32) is a critically important residue for basically all opioids (the authors cite one of the seminal studies demonstrating this). This residue, however, is part of the set used to distinguish ligands (e.g. part of PC2 in Figure 2F). This is reflected in the conclusions – the authors state that ligands with enhanced activity with depolarization interact with ECL2/TM6 while ligands with decreased activation with depolarization interact with TM3. However, this doesn't appear to be concordant with the docking poses presented in Figure 2 as morphine (enhanced activity) is shown to interact with Y148 in TM3.

Overall this study therefore provides interesting evidence that interactions between the MOR and the ligand in the binding pocket are important for MOR voltage sensitivity. But the specific interactions made and how they drive this behavior remains hard to rationalize based on the docking/fingerprint analysis

It is unclear why the authors initially present their fentanyl docking results as not being dependent on D147. This suggests that the docking needs further work and doesn't recapitulate key aspects of the binding site.

In Figure 1 – it is very hard to understand how/why morphine gives a negative response in the presence of a depolarizing current. This almost suggests there is a reserve of G protein that cannot interact with the receptor in the absence of depolarization. Some comment on this would be important.

In Figure 2 – it is unclear what the axes for PC1 and PC2 mean. Is a more positive value associated with increased reliance on this component?

In pg 7, the authors suggest that in silico modeling of mutations suggests that ligands bind to the mutated receptors with distinct poses, and this is responsible for the decreased EC50's observed in Figure 3. I think the authors need to be careful here – in general, it is far more likely that the binding pose of various ligands remains similar to the wild-type receptor. For example, it seems quite unlikely that the Y326F mutation would completely flip the pose for fentanyl in the binding site as it is known that D147 is important for fentanyl binding. Similarly, it is highly unlikely that methadone loses an ionic interaction in the H297A mutant.

Figure 4B is very challenging to interpret – it is unclear how the reader is able to figure out what is happening in the voltage dimension.

Figure 5E – it is unclear whether etorphine has any voltage dependent effect. It appears from this plot that it doesn't.

For the predictive experiments showing in Figure 5 – was etorphine blinded in the original analysis? It is unclear how this was a predictive result instead of another example of concordance between the fingerprint analysis and voltage sensitivity.

*Reviewer #3 (Recommendations for the authors):*

The authors combined electrophysiology with FRET assays to monitor how membrane potential modulates the ligand-dependent MOR signaling. The approach was described in their previous work [Ruland et al. Br J Pharmacol 2020, 177(15):3489-3504]. The current study focuses on how different receptor-ligand interactions are associated with differential voltage sensitivities of MOR signaling induced by various opioids. To determine the ligand binding modes, molecular docking and site-directed mutagenesis were performed. The authors proposed common receptor-ligand interaction patterns that may mechanically modulate MOR activation upon depolarization and alter Gαi signaling.

Voltage dependence may indeed be a non-negligible factor in the therapeutic outcome of opioids. Given the clinical importance of opioid analgesics, the large amount of data reported here are an asset in this largely untapped domain of opioid function. Structural insights into the ligand-specific voltage dependence of MOR would be valuable. However, the following aspects of the current manuscript raise concerns.

1. Accurate ligand binding mode prediction is critical for the conclusion drawn in this work. This can be challenging for ligands that are dissimilar to those in the crystal/cryo-EM structures (i.e. morphinan compounds, DAMGO, JDTic for opioid receptors). In addition, GPCR activation is a complex and dynamic allosteric process. A ligand may have multiple binding poses, and the active pose may not be the most energetically favorable. Even high-resolution experimental structures or elaborated molecular simulations can only partially describe the dynamic ligand-receptor interactions. Docking to a single crystal structure often results in irrelevant binding poses, such as the difficulties encountered with fentanyl in this work. While site-directed mutagenesis is commonly employed to verify ligand binding modes, often times it is not straightforward to interpret the data, due to the allosteric nature of GPCR activation dynamics. The approach in this study appears to perform well on morphinan ligands, for which the binding mode is known. The ability of the model to extrapolate to other scaffolds is questionable. For instance, the current study excluded DAMGO and SR17018 from the applicability domain. How to determine the applicability domain, if one were to use this approach for prospective predictions?

2. The fingerprint and principal component analyses further simplified the description of ligand-receptor interactions. It appears that the two categories of ligands (increasing or decreasing Gαi signaling upon depolarization) were better separated by the second principal component, which accounted for 14% of the variance in the interactions. To assess the model predictivity, the authors chose two morphinan ligands (etorphine and naloxone) as test cases. However, this does not reflect the model applicability to other ligand scaffolds, since the binding mode of morphinan ligands is known and the morphinan structure can be discriminated from other opioids. How predictive really is this seemingly over simplified model? Other type of ligands should be tested to provide better assessments.

3. Alternative test cases could be the large number of mutants reported here (Figure S6). Is the docking-fingerprint analysis able to predict the voltage sensitivity of the mutant receptors with the three ligands?

4. One major challenge here is to predict and verify ligand binding modes. The authors' choice of the PDB 5C1M (e.g. over 6DDE) was, understandably, to be more suitable for docking small molecular agonists. This made it difficult "to calculate a reliable binding mode for DAMGO due to the high flexibility of this peptidergic ligand". Even small molecules like fentanyl were difficult to predict. There is currently no easy solution. Molecular simulations have been reported to predict the binding mode of fentanyl, PZM21, TRV130, etc., which gave rather inconsistent results. Another possibility is ensemble docking which, however, is also non-trivial.

I think the key question is: how predictive is the final model, despite the simplifications of docking and principal component analysis? The manuscript as it currently stands, does not allow sufficient evaluation of this point. The authors may either challenge the model with non-morphinan ligands and the mutants, or provide a clearer distinction of the model's applicability domain.

5. How does the predicted fentanyl binding mode compare with lofentanyl in the recent cryo-EM structure (https://doi.org/10.1101/2021.12.07.471645)?

6. How does the apo receptor react to depolarization, if the signal is measurable?

7. Have about the other signaling pathways such as arrestins and GRK? Only some data are present for few ligands. Since the authors mentioned biased signaling in the introduction, one would expect the same analysis on these pathways. I am not suggesting to perform the additional experiments. However, the challenges should be discussed.

*Reviewer #4 (Recommendations for the authors):*

Here, the authors build on their prior work to investigate how the efficacy of various opioid agonists at the mu-opioid receptor (MOR) depends upon voltage (i.e. upon differences in voltage across the cell membrane). The authors employ a FRET-based screening assay that depends on proximity of the G-α and G-β subunits and monitor whole cell FRET under voltage clamping. First, the authors demonstrate that morphine, but not methadone or fentanyl, exhibits an increased ability to stimulate G-protein activation upon cell depolarization. Previously, the authors examined the voltage dependence of MOR activation by morphine, DAMGO, met-enkephalin and fentanyl (Ruland et al., 2020). Next, the authors used computational docking methods to predict the poses of various opioid ligands in the MOR binding pocket. The authors describe each pose in terms of an interaction 'fingerprint'--i.e. a list of all the residues in the binding pocket that interact with a chemical moiety on the ligand--and then selected residues to mutate to disrupt those interactions to test their binding predictions. Finally, the authors sought to link differences in interaction fingerprints for a given ligand with its voltage dependence, allowing for the identification of distinct sets of residues that engage ligands whose efficacy increases upon depolarization vs. those whose efficacy decreases upon depolarization. Overall, the work presented here could aid in the identification of candidate opioid compounds with certain physiological properties.

Strengths:

Pinpointing the atomic-level mechanisms underlying voltage sensitivity in a membrane protein receptor can be incredibly difficult. Thus, the authors take a chemo-informatics approach to try to correlate ligand-protein interactions with voltage dependence instead. A key strength of this paper is that this general approach could be used to derive predictions about other key properties that may be difficult to investigate using traditional mechanistic/biochemical approaches. Additionally, this same approach affords some predictive power, because one could imagine using the rules developed here to guide design of new ligands with certain voltage-dependent properties.

Weaknesses:

Although the manuscript presents, in principle, a sound strategy for examining a hard-to-study property of GPCRs, certain aspects of the experimental design require further support and validation to match the strength of the conclusions drawn.

First, a major component of the paper relies on the results of chemical docking predictions, which come with several caveats. Recent reports suggest that while docking software can generate numerous reasonable poses, often including the 'correct' pose, correctly ranking poses among a set of top poses remains challenging and may require manual selection of the pose based on chemical intuition. Although the authors do explain how they selected the 'final' pose for each of the predicted poses, it is not clear which other candidate poses might have been considered viable. While the authors use mutagenesis (see more below) to ablate key interactions between chemical groups on the ligand and receptor residues, they do not demonstrate the ability to rule out substantially different and incorrect poses using a similar mutagenesis strategy. On a similar note, although very few structures of MOR bound to different ligands have been experimentally determined, the authors should demonstrate the extent to which their docking approach is able to recover known ligand poses for those few experimentally determined structures. Additionally, they should map the interaction fingerprints resulting from the crystallographic poses into the principal components space shown in e.g. Figure 2F to reveal the similarity of known poses to docked poses for the same and different ligands.

A second concern stems from the authors' use of mutagenesis, coupled with a measure of receptor activation, as a primary way to validate binding pose predictions. In addition to the concern mentioned in the above paragraph, the authors use a FRET signal that reports on G-protein activation to assess ligand binding. Strictly speaking, one cannot uncouple determinants for binding from determinants for efficacy using this approach. Ideally, the authors would use radioligand binding/competition experiments to demonstrate the impact of mutating a residue on ligand binding, without the confounding variable of ligand efficacy.

Third, it is exciting to see that ligands that demonstrate either positive or negative activation due to depolarization can be differentiated by their interaction fingerprints with the receptor binding pocket. However, the various models described in the paper's conclusions do not afford an explanation for how voltage could differentially affect ligand binding (and/or activation) via these different surfaces. GPCRs are well known to be modulated by sodium and other ions, as well as by changes in receptor protonation. Without some clear mechanistic explanation of how voltage can impact the structure of the MOR in a ligand-dependent manner, the conclusions of this paper may have limited utility for understanding why GPCRs in general may be voltage sensitive, and how ligand activity may depend on this property.

[Editors’ note: further revisions were suggested prior to acceptance, as described below.]

Thank you for submitting the paper "Differential recognition of opioid analgesics by µ opioid receptors: Predicted interaction patterns correlate with ligand-specific voltage sensitivity" for consideration by *eLife*. Your article has been reviewed by 2 peer reviewers, and the evaluation has been overseen by a Reviewing Editor and a Senior Editor. The reviewers have opted to remain anonymous.

Comments to the Authors:

The editor and reviewers agree that your findings of ligand-dependent voltage sensitivity are intriguing, as is the observation about naloxone acting as a weak agonist under depolarizing conditions (although this also raises questions about physiological implications). However, the fundamental problem with the manuscript as written is the uncertainty about the accuracy of the docking poses. The mutant receptors really would need to be evaluated in a binding assay to look at this, not just a signaling assay. We think a substantially rewritten manuscript could make a more compelling story if it were to focus first and foremost on the experimental data and not emphasize the docking so much, given the impossibility of determining with certainty if the poses are accurate. In addition, now that additional structures of mu-OR bound to various ligands exist, we wonder whether the authors could take advantage of those structures (and MD simulation results described in Qu et al. Nat Chem Biol 2022) to guide mechanistic exploration of ligand-dependent voltage dependence, which would likely lead to a more compelling contribution to the field. If you decide to try and address these concerns with an extensively revised manuscript, we would be willing to consider it as a new submission. You would need to fully address the above concerns and all those detailed in the reviews below.

*Reviewer 1:*

In general, I am not certain that the authors have sufficiently addressed the concerns I raised previously:

1) Pose prediction is inherently incredibly difficult, particularly for the mu-opioid receptor, where the ligand binding pocket is flexible, and the chemical structure of ligands vary significantly. Consequently, while docking software can often generate several candidate poses that often include the 'true' pose, pose ranks may not be incredibly reliable. I would recommend that the authors, at a minimum, show for a subset of ligands what the alternative poses look like, and whether they all consistently make the same, or different contacts, within the binding pocket. (i.e. how much does the interaction fingerprint vary with pose?)

2) An additional, remaining concern is that EC50 measurements of G-protein activation carried out on different mutant receptors, which may have drastically different efficacies, do not directly validate binding pose. I am aware of only two primary ways to evaluate ligand binding directly: (a) radioligand binding/competition experiments or (b) determining the ligand-bound receptor structure, through X-ray crystallography or cryo-EM. I agree that performing either set of these experiments might be prohibitive for the lab and should not necessarily prevent publication certain results within this manuscript, which may be of broad interest to the GPCR community.

3) Related to the above point, even if we accept that G-protein activation is an acceptable, indirect measure of binding, the authors would still need to demonstrate that mutagenesis experiments enable them to discriminate between correct and incorrect poses. For example, they need to explicitly state the hypothesized effect of a given mutation based on the docked pose, and then demonstrate whether or not the interaction had the 'expected' effect. Otherwise, readers cannot assess whether validation via signaling is working. Although this information is present to some degree in the manuscript, it is incredibly difficult to know exactly what is going on because in Figure 2, not all binding pocket residues are shown within the binding pocket, even if they do not interact; and in Figure 3, the effects of some mutations are shown for some ligands, but not for others.

Perhaps the authors could carefully go through their manuscript to determine which conclusions are strongly supported by their data, and could re-organize their narrative to emphasize those findings. From my understanding, three key findings of significance that are well supported by the data are that:

1) The MOR displays different degrees of ligand-dependent voltage sensitivity across a wide array of agonists (Figure 4)

2) Certain residues in the binding pocket impact G-protein activation in a ligand-dependent manner in the absence of depolarization (Figure 3) and also in the presence of depolarization (Figure 5)--although as noted elsewhere, I do have concerns about how these data are currently presented & interpreted.

3) Naloxone, a known antagonist, (surprisingly) acts as a weak agonist under depolarizing conditions

The docking results are certainly interesting, and may be of use to the community, but need to be presented with the appropriate caveats and perhaps should not be the basis for all subsequent investigation in the manuscript.

Additional concerns:

1. In Figure 3D, I now think that the representation of change in EC50 through a simple ratio (and subsequent color coding) is quite difficult to interpret: e.g. a ratio of 500 indicates that the EC50 has shifted ~2 orders of magnitude to the right for mutant vs. WT. A ratio of 642 corresponds to a highly similar shift-yet this difference is represented as significant in the table, due to the color scheme. Would it be helpful to instead change this to a log (ratio) ?

2. It does not seem entirely correct to state that there is 'no correlation' between chemical structure and voltage sensitivity: indeed, buprenorphine, etorphine and morphine are all incredibly similar (share a morphinan scaffold), and they cluster together interns of their effects. (Figure S4K-L). Also, in Figure S4M, why do the interaction fingerprints (vertical columns) show all 0s for fentanyl and other similarly-behaved ligands?

3. In Figure 5, I do not understand why mutations to interacting residues e.g. Y148A and Y148F led to voltage-sensitive responses for morphine-this is just incredibly difficult to understand.

4. In Figure S2, I had intended for the DAMGO interaction fingerprint to be transformed into the space already described by the principal components analysis performed (i.e. no new PCA needs to be performed-one simply needs to figure out where DAMGO's fingerprint falls in the existing PC space, in order to figure out which DAMGO-MOR interactions are also common to interactions formed by the other ligands with MOR)

*Reviewer 2:*

The revised manuscript addresses many of the points raised in the initial round of review, however there are some substantial issues remaining that should, in my view, preclude publication of the manuscript in its current form. Most central is that the docking poses, which are critical to the entirety of the manuscript, remain incompletely evaluated. As noted in the first round of review, using G protein signaling assays with mutant receptors is an inherently confounded approach because G protein activation reflects both binding and signaling. Radioligand binding assays or another direct binding assay would be a more appropriate approach here, but the authors have chosen not to conduct these recommended experiments. A related concern is that expression levels of the mutants are not presented. If mutations affect expression levels, the EC50 value of the ligands may shift for this reason even without any direct change in binding affinity (radioligand assays would not be confounded in this way). Taken together, these shortcomings reduce confidence in the assigned binding poses which underpin the rest of the manuscript.

[Editors’ note: further revisions were suggested prior to acceptance, as described below.]

Thank you for resubmitting your work entitled "Differential recognition of opioid analgesics by µ opioid receptors: Predicted interaction patterns correlate with ligand-specific voltage sensitivity" for further consideration by *eLife*. Your revised article has been evaluated by Kenton Swartz (Senior Editor) and two reviewers.

The manuscript has been improved but there are some remaining issues that need to be addressed, as outlined below:

Although two of the reviewers who saw your earlier manuscript think the new revised manuscript is considerably improved, one of the reviewers would like to see data on the expression levels of the mutants to verify that they express to comparable levels.

*Reviewer #1 (Recommendations for the authors):*

The authors present a thorough and thoughtful revision of their manuscript, largely addressing prior reviewer critiques. Most importantly, they have effectively addressed concerns about docking poses with new direct binding data (although these data are still somewhat noisy) and with analysis of not only the top-scoring pose but now the top three poses. I have one significant concern remaining, which I believe should be addressed prior to publication.

Specifically, a variety of mutants are presented, but it appears these were evaluated only for DAMGO sensitivity in signaling and in competition assays, but not expression per se. A surface staining experiment or similar would be required to ensure that expression levels of mutants are not radically altered compared to WT. DAMGO activation is a proxy for expression and folding in some sense, but will also be influenced by mutation effects on DAMGO potency/efficacy, and might reflect pathway-saturated conditions where partial agonists would be sensitive to expression differences while full agonists like DAMGO are not. In general, evaluating receptor point mutants can be complex because they may affect expression, trafficking, ligand potency, and ligand efficacy. Direct measurements of many of these properties help to ensure correct interpretation of why mutations have the effects they do.

A variety of mutants are presented, but it appears these were evaluated only for DAMGO sensitivity in signaling and in competition assays, but not expression per se.

*Reviewer #2 (Recommendations for the authors):*

Overall, the paper seems to have improved, both in terms of presentation of results and high-level ideas and in terms of additional efforts made to directly characterize ligand-binding affinity via displacement assay.

One small recommendation is to avoid presenting main-text data as numerical tables in figures, e.g. especially in Figure 3D, I recommend presenting the data in perhaps a bar graph format, and using some sort of scheme to indicate the predicted vs. actual effect of the mutant on binding and activation (the latter may go in the paper's supplement).

---

## [Author Response]

[Editors’ note: the authors resubmitted a revised version of the paper for consideration. What follows is the authors’ response to the first round of review.]

Comments to the Authors:We are sorry to say that, after consultation with the reviewers, we have decided that this work will not be considered further for publication by eLife.Specifically, all three reviewers raised major concerns about the inherent uncertainty of the docking poses and the fingerprint analysis, which are central to the manuscript and its conclusions. While there may be some paths toward addressing this in principle (see comments below), the reviewers felt that the scope of needed work is beyond what would be reasonable for revision timeline. As a result, we cannot proceed with the current manuscript.Reviewer #2 (Recommendations for the authors):Over the past decade or so, a number of studies have demonstrated that multiple GPCRs are sensitive to voltage – i.e. their signaling properties change as a function of membrane voltage. The physiological consequence and relevance of this voltage sensitivity remains poorly understood. The authors recently reported that voltage can influence activation of the mu-opioid receptor, and that this effect is different between opioid receptor agonists (Ruland, BJP 2020), with some agonists causing increased activation of the receptor with depolarization, while other agonists (fentanyl) cause decreased activation with depolarization. In the present study, the authors aim to connect this observation to specific interactions made by different opioid receptor ligands to try to map specific contacts between ligands and the receptor that are responsible for this voltage sensitivity. Here, the authors first establish that different ligands acting at the opioid receptor are differentially sensitive to membrane voltage, e.g. the effect of morphine is increased while that of methadone and fentanyl is decreased. Based on these findings, the authors then dock various opioid ligands into the active conformation of MOR to identify differences in contacts between the ligands. A fingerprint analysis of contacts is used to cluster ligands with similar/different contacts; some of these differential interactions are probed/validated by dose response assays. The authors then suggest that grouping ligands with the fingerprint analysis approach seemed to correlate with voltage sensitivity, and that this correlation came with sufficient power to predict how meptazinol binds and the effect of voltage on etorphine. Specific mutations in the MOR binding pocket were then assessed for their effect on MOR voltage sensitivity to three prototypical opioids (morphien, methadone, and fentanyl) – here, the authors found that some mutations could reverse the directionality of voltage sensitivity for fentanyl and methadone. Finally, membrane depolarization could convert the antagonist naloxone into a weak partial agonist.These studies convincingly show that voltage does not simply change the conformation of the MOR, leading to more or less G protein signaling activity. Instead, it is the specific interactions of a ligand with the MOR binding pocket that lead to a specific outcome – these are clearly influenced by specific ligands, and the specific contacts that a given ligand makes with the MOR binding pocket. The most exciting data to the latter point is the remarkable effect that specific mutations can have in converting the effect of voltage on methadone in Figure 6C or in the effect on naloxone.Less clear is how well the authors can define these specific interactions. It is not clear whether the docking studies are truly informative in this regard. First, but it is not completely clear whether binding poses for various ligands are truly accurate – the authors efforts to dock fentanyl suggest that there are many potential challenges in obtaining accurate poses. Even with accurate poses, it is unclear whether the fingerprint analysis is robust. For example, it is likely the case that D147(3.32) is a critically important residue for basically all opioids (the authors cite one of the seminal studies demonstrating this). This residue, however, is part of the set used to distinguish ligands (e.g. part of PC2 in Figure 2F). This is reflected in the conclusions – the authors state that ligands with enhanced activity with depolarization interact with ECL2/TM6 while ligands with decreased activation with depolarization interact with TM3. However, this doesn't appear to be concordant with the docking poses presented in Figure 2 as morphine (enhanced activity) is shown to interact with Y148 in TM3.Overall this study therefore provides interesting evidence that interactions between the MOR and the ligand in the binding pocket are important for MOR voltage sensitivity. But the specific interactions made and how they drive this behavior remains hard to rationalize based on the docking/fingerprint analysisIt is unclear why the authors initially present their fentanyl docking results as not being dependent on D147. This suggests that the docking needs further work and doesn't recapitulate key aspects of the binding site.

We thank the reviewer for bringing up this important point. We have re-performed the docking and all following analysis. For the docking, we set D147 as mandatory interaction. Further, for fentanyl we used for our analysis the recently published structure of the MOR-fentanyl complex (PDB: 8EFS, Zhuang et al., 2022), as our binding pose was flipped upside down in comparison to this structure. This had no influence on the overall interactions which we calculated. This could be due to the inherent symmetry in fentanyl, making the prediction of a general binding mode for fentanyl in general difficult.

In Figure 1 – it is very hard to understand how/why morphine gives a negative response in the presence of a depolarizing current. This almost suggests there is a reserve of G protein that cannot interact with the receptor in the absence of depolarization. Some comment on this would be important.

In Figure 1 we used a FRET-based G-protein activation sensor published by van Unen et al. 2016. This sensor has a fluorescence labeled Gai and Gγ. Upon activation of the receptor, the G-proteins are activated and dissociate or reorient, resulting in a decrease in the FRET emission ratio. With this, a decrease, like seen for morphine, indicates an activation. Further, we used non-saturating concentrations of the respective agonists, resulting in a reserve of G-proteins which are not activated through the application of agonist, but are indeed activated the increased activity of the receptor bound to morphine under depolarized conditions.

In Figure 2 – it is unclear what the axes for PC1 and PC2 mean. Is a more positive value associated with increased reliance on this component?

Principal Components 1 and 2 (PC1 and PC2) represent linear combinations of interaction fingerprints that explain the largest variance in the interaction fingerprints. PC1 and PC2 themselves do not carry any meaning except to represent distances between docking poses based on their interaction fingerprints. We added the following line in the material and methods to make it clearer: “Values on the x- and y-axis, respectively, originate from the linear combination of fingerprint features and do not carry an additional meaning, e.g. likelihood”.

In pg 7, the authors suggest that in silico modeling of mutations suggests that ligands bind to the mutated receptors with distinct poses, and this is responsible for the decreased EC50's observed in Figure 3. I think the authors need to be careful here – in general, it is far more likely that the binding pose of various ligands remains similar to the wild-type receptor. For example, it seems quite unlikely that the Y326F mutation would completely flip the pose for fentanyl in the binding site as it is known that D147 is important for fentanyl binding. Similarly, it is highly unlikely that methadone loses an ionic interaction in the H297A mutant.

We appreciate the comments and agree with the reviewer to be more cautious here. For this reason, we excluded the docking to the mutated receptor from the present manuscript.

Figure 4B is very challenging to interpret – it is unclear how the reader is able to figure out what is happening in the voltage dimension.

We have redone Figure 4B in order to hopefully make it clearer.

Figure 5E – it is unclear whether etorphine has any voltage dependent effect. It appears from this plot that it doesn't.

We have excluded Figure 5 from the manuscript. As we had to reperform all docking calculations, we cannot make predictions anymore. Furthermore, we just see a very small voltage-induced activation under etorphine, which is significantly smaller than the effect induced my morphine (see Figure 4a)

For the predictive experiments showing in Figure 5 – was etorphine blinded in the original analysis? It is unclear how this was a predictive result instead of another example of concordance between the fingerprint analysis and voltage sensitivity.

As we reperformed the docking calculations, we do not present any predictive results in the manuscript anymore and we excluded Figure 5.

Reviewer #3 (Recommendations for the authors):The authors combined electrophysiology with FRET assays to monitor how membrane potential modulates the ligand-dependent MOR signaling. The approach was described in their previous work [Ruland et al. Br J Pharmacol 2020, 177(15):3489-3504]. The current study focuses on how different receptor-ligand interactions are associated with differential voltage sensitivities of MOR signaling induced by various opioids. To determine the ligand binding modes, molecular docking and site-directed mutagenesis were performed. The authors proposed common receptor-ligand interaction patterns that may mechanically modulate MOR activation upon depolarization and alter Gαi signaling.Voltage dependence may indeed be a non-negligible factor in the therapeutic outcome of opioids. Given the clinical importance of opioid analgesics, the large amount of data reported here are an asset in this largely untapped domain of opioid function. Structural insights into the ligand-specific voltage dependence of MOR would be valuable. However, the following aspects of the current manuscript raise major concerns.1. Accurate ligand binding mode prediction is critical for the conclusion drawn in this work. This can be challenging for ligands that are dissimilar to those in the crystal/cryo-EM structures (i.e. morphinan compounds, DAMGO, JDTic for opioid receptors). In addition, GPCR activation is a complex and dynamic allosteric process. A ligand may have multiple binding poses, and the active pose may not be the most energetically favorable. Even high-resolution experimental structures or elaborated molecular simulations can only partially describe the dynamic ligand-receptor interactions. Docking to a single crystal structure often results in irrelevant binding poses, such as the difficulties encountered with fentanyl in this work. While site-directed mutagenesis is commonly employed to verify ligand binding modes, often times it is not straightforward to interpret the data, due to the allosteric nature of GPCR activation dynamics. The approach in this study appears to perform well on morphinan ligands, for which the binding mode is known. The ability of the model to extrapolate to other scaffolds is questionable. For instance, the current study excluded DAMGO and SR17018 from the applicability domain. How to determine the applicability domain, if one were to use this approach for prospective predictions?

We like to thank the review for bringing up this point, we now included SR17018 in the analysis. The reviewer is right that our findings may not extend to predictions of arbitrary ligands, and as we are not performing systematic predictions in this manuscript, this is difficult to quantitatively assess. Still, based on our observations during the revisions to this manuscript, we feel confident that we are able to give the reader cautionary advice, and therefore added the following line in page 5 under Results: “Along these lines, we suggest that the use of our findings in a predictive manner should only be attempted for ligands with similar physicochemical characteristics (including the size; Table S1) and binding locations.”

2. The fingerprint and principal component analyses further simplified the description of ligand-receptor interactions. It appears that the two categories of ligands (increasing or decreasing Gαi signaling upon depolarization) were better separated by the second principal component, which accounted for 14% of the variance in the interactions. To assess the model predictivity, the authors chose two morphinan ligands (etorphine and naloxone) as test cases. However, this does not reflect the model applicability to other ligand scaffolds, since the binding mode of morphinan ligands is known and the morphinan structure can be discriminated from other opioids. How predictive really is this seemingly over simplified model? Other type of ligands should be tested to provide better assessments.

As mentioned in our reply to the previous point 1, we think that a systematic analysis of this question is beyond the scope of this manuscript. Moreover, we did not really intend to suggest to use our findings in a predictive manner. Of course, if our hypotheses are correct, this should be possible, but we tried to carefully rephrase the text so that we do not make this claim.

3. Alternative test cases could be the large number of mutants reported here (Figure S6). Is the docking-fingerprint analysis able to predict the voltage sensitivity of the mutant receptors with the three ligands?

The docking calculations to the mutant receptors are complicated due to the sensitivity of docking towards slight changes in the binding pocket. Therefore, we decided to exclude the docking to mutant receptors, especially since we do not have the crystallographic structure of the mutants available and can therefore not assess the validity of our computational predictions.

4. One major challenge here is to predict and verify ligand binding modes. The authors' choice of the PDB 5C1M (e.g. over 6DDE) was, understandably, to be more suitable for docking small molecular agonists. This made it difficult "to calculate a reliable binding mode for DAMGO due to the high flexibility of this peptidergic ligand". Even small molecules like fentanyl were difficult to predict. There is currently no easy solution. Molecular simulations have been reported to predict the binding mode of fentanyl, PZM21, TRV130, etc., which gave rather inconsistent results. Another possibility is ensemble docking which, however, is also non-trivial.I think the key question is: how predictive is the final model, despite the simplifications of docking and principal component analysis? The manuscript as it currently stands, does not allow sufficient evaluation of this point. The authors may either challenge the model with non-morphinan ligands and the mutants, or provide a clearer distinction of the model's applicability domain.

We understand the reviewer’s concern with regards to the applicability and have added the following line in page 5 under Results: “Along these lines, we suggest that the use of our findings in a predictive manner should only be attempted for ligands with similar physicochemical characteristics (including the size; Table S1) and binding locations.”

5. How does the predicted fentanyl binding mode compare with lofentanyl in the recent cryo-EM structure (https://doi.org/10.1101/2021.12.07.471645)?

In the meantime, a structure of the MOR-fentanyl complex was published, on which we based our fentanyl analysis (PDB: 8EFS, Zhuang et al., 2022), as our binding pose was flipped upside down in comparison to this structure.

6. How does the apo receptor react to depolarization, if the signal is measurable?

This can be found in Supplemental Figure 1a. In the apo receptor, without the application of agonist, there is no effect due to depolarization.

7. Have about the other signaling pathways such as arrestins and GRK? Only some data are present for few ligands. Since the authors mentioned biased signaling in the introduction, one would expect the same analysis on these pathways. I am not suggesting to perform the additional experiments. However, the challenges should be discussed.

We have removed the biased signaling in the introduction. As we worked with several partial agonists, the measurement in GRK or Arrestin assays is not simple. We tried to perform the assays, but they did not really work.

Reviewer #4 (Recommendations for the authors):Here, the authors build on their prior work to investigate how the efficacy of various opioid agonists at the mu-opioid receptor (MOR) depends upon voltage (i.e. upon differences in voltage across the cell membrane). The authors employ a FRET-based screening assay that depends on proximity of the G-α and G-β subunits and monitor whole cell FRET under voltage clamping. First, the authors demonstrate that morphine, but not methadone or fentanyl, exhibits an increased ability to stimulate G-protein activation upon cell depolarization. Previously, the authors examined the voltage dependence of MOR activation by morphine, DAMGO, met-enkephalin and fentanyl (Ruland et al., 2020). Next, the authors used computational docking methods to predict the poses of various opioid ligands in the MOR binding pocket. The authors describe each pose in terms of an interaction 'fingerprint'--i.e. a list of all the residues in the binding pocket that interact with a chemical moiety on the ligand--and then selected residues to mutate to disrupt those interactions to test their binding predictions. Finally, the authors sought to link differences in interaction fingerprints for a given ligand with its voltage dependence, allowing for the identification of distinct sets of residues that engage ligands whose efficacy increases upon depolarization vs. those whose efficacy decreases upon depolarization. Overall, the work presented here could aid in the identification of candidate opioid compounds with certain physiological properties.Strengths:Pinpointing the atomic-level mechanisms underlying voltage sensitivity in a membrane protein receptor can be incredibly difficult. Thus, the authors take a chemo-informatics approach to try to correlate ligand-protein interactions with voltage dependence instead. A key strength of this paper is that this general approach could be used to derive predictions about other key properties that may be difficult to investigate using traditional mechanistic/biochemical approaches. Additionally, this same approach affords some predictive power, because one could imagine using the rules developed here to guide design of new ligands with certain voltage-dependent properties.Weaknesses:Although the manuscript presents, in principle, a sound strategy for examining a hard-to-study property of GPCRs, certain aspects of the experimental design require further support and validation to match the strength of the conclusions drawn.First, a major component of the paper relies on the results of chemical docking predictions, which come with several caveats. Recent reports suggest that while docking software can generate numerous reasonable poses, often including the 'correct' pose, correctly ranking poses among a set of top poses remains challenging and may require manual selection of the pose based on chemical intuition. Although the authors do explain how they selected the 'final' pose for each of the predicted poses, it is not clear which other candidate poses might have been considered viable. While the authors use mutagenesis (see more below) to ablate key interactions between chemical groups on the ligand and receptor residues, they do not demonstrate the ability to rule out substantially different and incorrect poses using a similar mutagenesis strategy. On a similar note, although very few structures of MOR bound to different ligands have been experimentally determined, the authors should demonstrate the extent to which their docking approach is able to recover known ligand poses for those few experimentally determined structures. Additionally, they should map the interaction fingerprints resulting from the crystallographic poses into the principal components space shown in e.g. Figure 2F to reveal the similarity of known poses to docked poses for the same and different ligands.

We would like to thank the reviewer for their concern. We re-did the docking and only selected the best scored pose for each ligand with the exception of Pethidine. We also compared the docking pose of fentanyl to the one in the recently published structure of MOR (PDB 8EF5) and found that despite the binding pose being flipped, the overall interactions were comparable. This is due to the high inherent symmetry of fentanyl. We added the analysis of the previously calculated docking pose in the Results section.

A second concern stems from the authors' use of mutagenesis, coupled with a measure of receptor activation, as a primary way to validate binding pose predictions. In addition to the concern mentioned in the above paragraph, the authors use a FRET signal that reports on G-protein activation to assess ligand binding. Strictly speaking, one cannot uncouple determinants for binding from determinants for efficacy using this approach. Ideally, the authors would use radioligand binding/competition experiments to demonstrate the impact of mutating a residue on ligand binding, without the confounding variable of ligand efficacy.

We thank the reviewer for bringing up this important point. However, the main conclusions of this work deal with the voltage sensitivity. Radioligand binding or competition experiments are not practicable in this regard.

Third, it is exciting to see that ligands that demonstrate either positive or negative activation due to depolarization can be differentiated by their interaction fingerprints with the receptor binding pocket. However, the various models described in the paper's conclusions do not afford an explanation for how voltage could differentially affect ligand binding (and/or activation) via these different surfaces. GPCRs are well known to be modulated by sodium and other ions, as well as by changes in receptor protonation. Without some clear mechanistic explanation of how voltage can impact the structure of the MOR in a ligand-dependent manner, the conclusions of this paper may have limited utility for understanding why GPCRs in general may be voltage sensitive, and how ligand activity may depend on this property.

We cannot give an accurate mechanism for the impact of voltage on the overall structure of the MOR with these results. Our identified residues important for voltage sensitivity of the MOR are for example not known to be part of the GPCR activation pathways which where described elsewhere (see Hauser et al. 2021). It was also shown before that for example the sodium ion is not part of the voltage sensing mechanisms of GPCRs (see Ågren et al., 2018 and Tauber & Chaim, 2022).

[Editors’ note: what follows is the authors’ response to the second round of review.]

Reviewer 1:In general, I am not certain that the authors have sufficiently addressed the concerns I raised previously:1) Pose prediction is inherently incredibly difficult, particularly for the mu-opioid receptor, where the ligand binding pocket is flexible, and the chemical structure of ligands vary significantly. Consequently, while docking software can often generate several candidate poses that often include the 'true' pose, pose ranks may not be incredibly reliable. I would recommend that the authors, at a minimum, show for a subset of ligands what the alternative poses look like, and whether they all consistently make the same, or different contacts, within the binding pocket. (i.e. how much does the interaction fingerprint vary with pose?)

We appreciate the reviewer for bringing up this point. We repeated our fingerprint analysis for all tested ligands with not only the highest ranked poses but also with the top three poses according to energy score (Supplemental Figure 3B). The resulting fingerprints did not vary to a large extent between the top three poses, suggesting our computational pose prediction is suitable for further evaluation.

2) An additional, remaining concern is that EC50 measurements of G-protein activation carried out on different mutant receptors, which may have drastically different efficacies, do not directly validate binding pose. I am aware of only two primary ways to evaluate ligand binding directly: (a) radioligand binding/competition experiments or (b) determining the ligand-bound receptor structure, through X-ray crystallography or cryo-EM. I agree that performing either set of these experiments might be prohibitive for the lab and should not necessarily prevent publication certain results within this manuscript, which may be of broad interest to the GPCR community.

We would like to thank the reviewer for bringing up this important point again. We have now performed fluorescent ligand competition experiments for all of our mutants with the agonists morphine, methadone and fentanyl and calculated IC50 values for the respective agonists (figure 3B-D, supplemental figure 5-6). The results are in line with our previous evaluation of the mutant receptors, supporting our calculated interactions between ligand and receptor.

3) Related to the above point, even if we accept that G-protein activation is an acceptable, indirect measure of binding, the authors would still need to demonstrate that mutagenesis experiments enable them to discriminate between correct and incorrect poses. For example, they need to explicitly state the hypothesized effect of a given mutation based on the docked pose, and then demonstrate whether or not the interaction had the 'expected' effect. Otherwise, readers cannot assess whether validation via signaling is working. Although this information is present to some degree in the manuscript, it is incredibly difficult to know exactly what is going on because in Figure 2, not all binding pocket residues are shown within the binding pocket, even if they do not interact; and in Figure 3, the effects of some mutations are shown for some ligands, but not for others.

As mentioned above, we have now measured the binding directly via fluorescent ligand competition experiments. Further, we have reduced our statements to the point that we calculated interactions which we have confirmed via binding and signaling experiments. In addition, we have added 2D interaction maps displaying the calculated interactions (Figure 2D-G), which should simplify the understanding of our statements.

Perhaps the authors could carefully go through their manuscript to determine which conclusions are strongly supported by their data, and could re-organize their narrative to emphasize those findings. From my understanding, three key findings of significance that are well supported by the data are that:1) The MOR displays different degrees of ligand-dependent voltage sensitivity across a wide array of agonists (Figure 4)2) Certain residues in the binding pocket impact G-protein activation in a ligand-dependent manner in the absence of depolarization (Figure 3) and also in the presence of depolarization (Figure 5)--although as noted elsewhere, I do have concerns about how these data are currently presented & interpreted.3) Naloxone, a known antagonist, (surprisingly) acts as a weak agonist under depolarizing conditionsThe docking results are certainly interesting, and may be of use to the community, but need to be presented with the appropriate caveats and perhaps should not be the basis for all subsequent investigation in the manuscript.

We appreciate this comment, but as we present the data in this study is how we were performing all analysis: we observed the ligand specific voltage sensitivity of the MOR (as published before, Ruland et al. 2020). We next performed docking calculations and out of these calculations we searched for possible important interactions and decided to mutate these different residues in the ligand binding pocket.

Additional concerns:1. In Figure 3D, I now think that the representation of change in EC50 through a simple ratio (and subsequent color coding) is quite difficult to interpret: e.g. a ratio of 500 indicates that the EC50 has shifted ~2 orders of magnitude to the right for mutant vs. WT. A ratio of 642 corresponds to a highly similar shift-yet this difference is represented as significant in the table, due to the color scheme. Would it be helpful to instead change this to a log (ratio) ?

We thank the reviewer for bringing up this point, we have changed the EC50 to pEC50, and pIC50 respectively.

2. It does not seem entirely correct to state that there is 'no correlation' between chemical structure and voltage sensitivity: indeed, buprenorphine, etorphine and morphine are all incredibly similar (share a morphinan scaffold), and they cluster together interns of their effects. (Figure S4K-L). Also, in Figure S4M, why do the interaction fingerprints (vertical columns) show all 0s for fentanyl and other similarly-behaved ligands?

Our analysis in supplemental Figure 7 K-L revealed no correlation between the ligands. But indeed, it is correct that al morphinan ligands behave the same way, but also ligands with a completely different scaffold show the same effect (like pethidine and PZM21).

3. In Figure 5, I do not understand why mutations to interacting residues e.g. Y148A and Y148F led to voltage-sensitive responses for morphine-this is just incredibly difficult to understand.

We showed that these to mutants still enable binding of morphine and induce Gprotein activation (Figure 3 and Supplemental Figure 4-6), but the EC50 is shifted. For this, we increased the dose of morphine for the measurements shown in Figure 5A. Here we can still detect a morphine induced G-protein activation which is also sensitive for voltage changes. Comparable to WT, the G-protein activation is increased upon depolarization, but the voltage induced increase in activation is smaller than for WT (Y148F, Figure 5A green trace) or increased in comparison to WT (Y148A, pink trace). Overall, all mutants still led to a voltage-sensitive response for morphine, as for all mutants the morphine induced G-protein activation was increased upon depolarization.

4. In Figure S2, I had intended for the DAMGO interaction fingerprint to be transformed into the space already described by the principal components analysis performed (i.e. no new PCA needs to be performed-one simply needs to figure out where DAMGO's fingerprint falls in the existing PC space, in order to figure out which DAMGO-MOR interactions are also common to interactions formed by the other ligands with MOR)

We have now performed this analysis as shown in Supplemental Figure 3D. DAMGO does in this analysis still nut cluster with any of the other ligands, confirming the substantially different behavior of DAMGO in the docking and fingerprint analysis.

Reviewer 2:The revised manuscript addresses many of the points raised in the initial round of review, however there are some substantial issues remaining that should, in my view, preclude publication of the manuscript in its current form. Most central is that the docking poses, which are critical to the entirety of the manuscript, remain incompletely evaluated. As noted in the first round of review, using G protein signaling assays with mutant receptors is an inherently confounded approach because G protein activation reflects both binding and signaling. Radioligand binding assays or another direct binding assay would be a more appropriate approach here, but the authors have chosen not to conduct these recommended experiments. A related concern is that expression levels of the mutants are not presented. If mutations affect expression levels, the EC50 value of the ligands may shift for this reason even without any direct change in binding affinity (radioligand assays would not be confounded in this way). Taken together, these shortcomings reduce confidence in the assigned binding poses which underpin the rest of the manuscript.

We thank the reviewer for bringing up this point, we have, as mentioned above, now performed direct binding assays using fluorescent ligand competition experiments, which are in line with our previous results. Using the fluorescent ligand, we also could show that the mutant receptors are expressed in comparable levels as the WT (Supplemental Figure 5 T). All mutants which were able to bind the fluorescent ligand showed no significant different maximum intensities in comparison to WT, suggesting they are expressed in comparable levels. For the mutants which did not bind the fluorescent ligand anymore, we cannot make any statements regarding expression levels.

[Editors’ note: what follows is the authors’ response to the third round of review.]

The manuscript has been improved but there are some remaining issues that need to be addressed, as outlined below:Although two of the reviewers who saw your earlier manuscript think the new revised manuscript is considerably improved, one of the reviewers would like to see data on the expression levels of the mutants to verify that they express to comparable levels. The reviewers also have several minor comments that should be addressed.Reviewer #1 (Recommendations for the authors):The authors present a thorough and thoughtful revision of their manuscript, largely addressing prior reviewer critiques. Most importantly, they have effectively addressed concerns about docking poses with new direct binding data (although these data are still somewhat noisy) and with analysis of not only the top-scoring pose but now the top three poses. I have one significant concern remaining, which I believe should be addressed prior to publication.Specifically, a variety of mutants are presented, but it appears these were evaluated only for DAMGO sensitivity in signaling and in competition assays, but not expression per se. A surface staining experiment or similar would be required to ensure that expression levels of mutants are not radically altered compared to WT. DAMGO activation is a proxy for expression and folding in some sense, but will also be influenced by mutation effects on DAMGO potency/efficacy, and might reflect pathway-saturated conditions where partial agonists would be sensitive to expression differences while full agonists like DAMGO are not. In general, evaluating receptor point mutants can be complex because they may affect expression, trafficking, ligand potency, and ligand efficacy. Direct measurements of many of these properties help to ensure correct interpretation of why mutations have the effects they do.A variety of mutants are presented, but it appears these were evaluated only for DAMGO sensitivity in signaling and in competition assays, but not expression per se.

We would like to thank the editor and reviewers again for their critical remarks.

As reviewer 1 pointed out, we did not evaluate the expression levels of the receptor mutants per se. To address this, we performed western blot analysis with all mutants and compared their expression levels with the expression of the WT receptor. With this, we could confirm that all the mutants were expressed in comparable levels as the WT receptor. You can find these updated results in Figure 3 —figure supplement 4

Reviewer #2 (Recommendations for the authors):Overall, the paper seems to have improved, both in terms of presentation of results and high-level ideas and in terms of additional efforts made to directly characterize ligand-binding affinity via displacement assay.One small recommendation is to avoid presenting main-text data as numerical tables in figures, e.g. especially in Figure 3D, I recommend presenting the data in perhaps a bar graph format, and using some sort of scheme to indicate the predicted vs. actual effect of the mutant on binding and activation (the latter may go in the paper's supplement).

We appreciate this comment. We have revised Figure 3D (page 10), and the data is no presented in bar graphs now representing the effect of the mutant on binding and activation. The numerical data can be found in Supplementary file 2.